# A neural network for information seeking

J. Kael White[1,6], Ethan S. Bromberg-Martin [1,6], Sarah R. Heilbronner[2], Kaining Zhang[3], Julia Pai[1,4], Suzanne N. Haber[5] & Ilya E. Monosov[1,3]*

Humans and other animals often show a strong desire to know the uncertain rewards their future has in store, even when they cannot use this information to influence the outcome. However, it is unknown how the brain predicts opportunities to gain information and motivates this information-seeking behavior. Here we show that neurons in a network of interconnected subregions of primate anterior cingulate cortex and basal ganglia predict the moment of gaining information about uncertain rewards. Spontaneous increases in their information prediction signals are followed by gaze shifts toward objects associated with resolving uncertainty, and pharmacologically disrupting this network reduces the motivation to seek information. These findings demonstrate a cortico-basal ganglia mechanism responsible for motivating actions to resolve uncertainty by seeking knowledge about the future.

[1] Department of Neuroscience, Washington University School of Medicine, St. Louis, MO, USA. [2] Department of Neuroscience, University of Minnesota, Minneapolis, MN, USA. [3] Department of Biomedical Engineering, Washington University, St. Louis, MO, USA. [4] Center for Neural Science, New York University, New York, NY, USA. [5] Department of Pharmacology and Physiology, University of Rochester, Rochester, NY, USA. [6]These authors contributed equally: J. Kael White, Ethan S. Bromberg-Martin. *email: monosovi@wustl.edu

Humans and other animals often express a desire to know the uncertain rewards their future has in store, even when they cannot use this information to influence the outcome[1,2]. This information-seeking behavior is not predicted by standard theories of reinforcement learning and reward seeking, and has been the subject of investigation in the fields of psychology, economics, and neuroscience[3–8]. These studies have revealed neuronal populations suitable for regulating specific aspects of information-seeking behavior. Certain prefrontal cortical neurons in monkeys and areas in humans are transiently activated by visual cues associated with future information gain about uncertain outcomes[9,10]. In addition, evidence from both monkeys and humans suggests that gaining information about uncertain outcomes can be a form of reward, since it activates the same reward-prediction error circuitry as primary rewards like food and water[6,10–12].

These findings, however, leave a major question unanswered: how does the brain motivate ongoing information-seeking behavior? That is, how does the brain bridge the gap to sustain our motivation to seek information, in between the time when we first learn of an opportunity to gain information and the time when we finally obtain it? At present, it is completely unknown what neural networks are causally responsible for information-seeking behavior, and it is unknown what neural code they could use to regulate the strength of information seeking.

To gain insight into this question, we took inspiration from a field of study that has performed extensive investigations into the analogous question for conventional primary reward seeking. These studies have revealed that the brain contains neural populations encoding reward predictions. Many such neurons have sustained activity that starts from the moment when reward can first be predicted, scales with the expected amount of reward, and ramps up to the expected time when the reward become available[13]. These reward-prediction signals are linked to reward-seeking behavior: they correlate with reward seeking[14–17], and perturbing them alters reward seeking[18–20].

We therefore hypothesized that information seeking is motivated by an analogous neural network encoding information predictions. In order for a neural network to motivate ongoing information seeking, it must (A) monitor the level of uncertainty about future events, (B) anticipate the time when information will become available to resolve the uncertainty, (C) activate before information-seeking behaviors, such as gaze shifts to inspect the source of uncertainty, (D) causally motivate behavior to obtain information.

Here, we demonstrate that these criteria are met by an anatomically interconnected network comprising three areas of the primate brain: the anterior cingulate cortex (ACC) and two subregions of the basal ganglia (BG), the internal-capsule-bordering portion of the dorsal striatum (icbDS) and the anterior pallidum, including anterior globus pallidus and the ventral pallidum (Pal).

## Results

**A cortico-basal ganglia network monitors reward uncertainty.** To identify neurons selectively responsive to reward uncertainty, we presented monkeys with fractal visual conditioned stimuli (CSs) predicting future juice reward with 0%, 25%, 50%, 75%, and 100% probabilities[21,22]. All three areas contained numerous neurons that were strongly activated or inhibited by CSs that cued uncertain rewards (Fig. 1b; Fig. 1c, 25%, 50%, and 75% reward CSs). These responses were primarily excitatory in ACC and icbDS and often inhibitory in Pal (Fig. 1b). The average responses were sustained ramping to the moment the uncertain outcome would occur (Fig. 1c, d). Importantly, unlike conventional

reward-related neurons in these areas[21–23], these neurons were more responsive to reward uncertainty than reward value: their responses were substantially lower for the CSs that cued certain reward or certain no reward, even though they had the most extreme values (Fig. 1c–e, 100 and 0% reward CSs).

Furthermore, many of these neurons responded to uncertainty in a graded manner:[24] they responded most in the condition with maximal uncertainty (50% reward), less in conditions with intermediate uncertainty (75 and 25% reward), and least in conditions with no uncertainty (100 and 0% reward)[21,22]. Specifically, all three areas had neurons with a significantly different average response to the 50% CS than the 75 and 25% CSs (Fig. 1f, $p < 0.05$, signed-rank test) at greater prevalence than expected by chance (19% of neurons ($n = 32/165$); $p < 0.001$ in each area, binomial tests). Of these neurons, 97% had significantly stronger responses to maximal uncertainty than to intermediate uncertainty, consistent with graded uncertainty coding, while only 3% had the opposite pattern (Fig. 1f, $n = 31$ vs. $n = 1$). As a result, there were significantly more neurons with responses consistent with graded uncertainty coding than neurons with the opposite pattern (Fig. 1f; $p < 0.001$ in each area, binomial tests) and the net differential activity was greater for maximal uncertainty than intermediate uncertainty ($p < 0.01$ in each area, signed-rank tests). Further tests confirmed that these areas had the hallmarks of graded uncertainty coding (Supplementary Note 1).

The network refined its signals over time. Uncertainty signals emerged markedly earlier in Pal, but this initial signal did not predominantly encode the graded level of uncertainty (Fig. 1d, arrows; significantly shorter latency in Pal than ACC and icbDS, both $p \leq 0.005$, permutation tests; Supplementary Fig. 1). Uncertainty signals later emerged in ACC and icbDS at similar latencies (Fig. 1d; no significant latency difference, $p = 0.35$), and these areas first significantly encoded the graded level of uncertainty (50 > 25, 75%, Fig. 1d; Supplementary Fig. 1). Thus, a rapid but rough Pal uncertainty signal was followed by a slower, graded signal in cortico-striatal areas.

Given the closely related uncertainty signals in these areas, we tested whether they form an interconnected network. We injected the bidirectional tracer Lucifer yellow (LY) into the internal-capsule-bordering regions of DS where uncertainty-responsive neurons were found (Fig. 1a; Supplementary Fig. 2). This produced a large number of retrogradely labeled cells in the ACC (Fig. 1a), including the subregion containing uncertainty-responsive neurons (Fig. 1b), indicating a unidirectional ACC→icbDS projection. In addition, this produced both retrogradely labeled cells and anterogradely labeled fibers in Pal (Fig. 1a), including the region containing uncertainty-responsive neurons (Fig. 1b), indicating bidirectional icbDS→Pal and Pal→icbDS projections. Bidirectional icbDS→Pal and Pal→icbDS projections were confirmed by tracer injection into Pal (Supplementary Fig. 2). These connections are consistent with established cortico-BG circuits[25–27], and are ideally suited to support the uncertainty coding we observed. Notably, these connections are consistent with uncertainty coding being primarily excitatory in ACC and icbDS and more commonly inhibitory in Pal (Fig. 1b; Supplementary S1) since cortex–striatum projections are excitatory and striatum–pallidum projections are mutually inhibitory[28]. They are also consistent with uncertainty signals emerging first in Pal, as Pal can communicate with ACC and icbDS via direct Pal→icbDS projections and classic cortex→Pal→thalamus→cortex loops[25].

**The network anticipates information to resolve uncertainty.** Our findings identify a cortico-BG network that signals reward

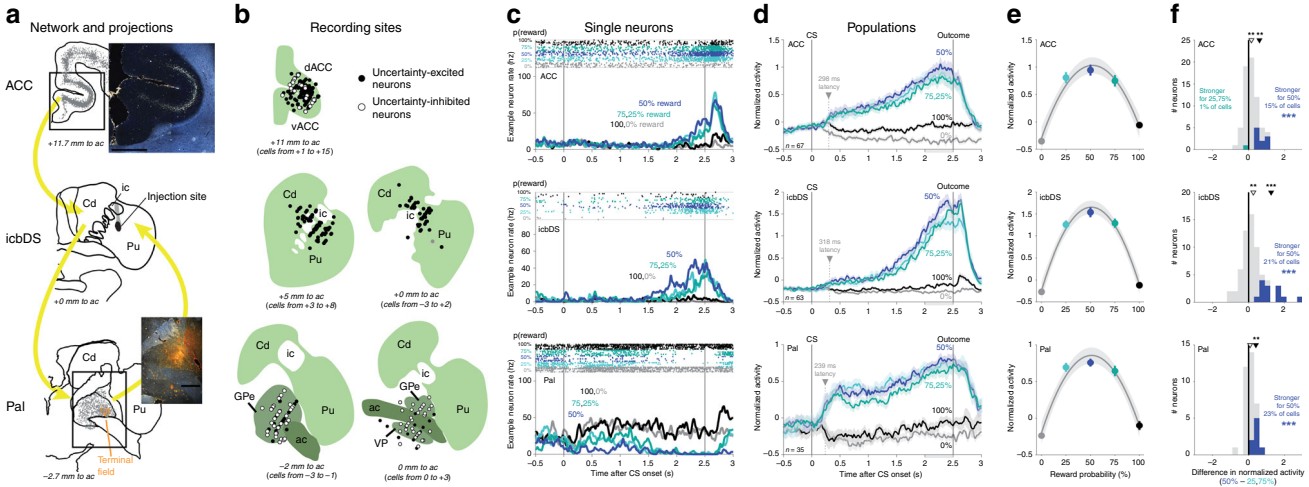

**Fig. 1** A cortico-basal ganglia network signals reward uncertainty. **a** ACC–icbDS–Pal network. Lucifer yellow was injected in icbDS (middle, black area). Retrogradely labeled cell bodies (gray) were found in ACC (top; inset) and Pal (bottom). Anterogradely labeled fibers (orange area) were found in Pal (bottom; inset). ic: internal capsule, Cd: caudate nucleus, Pu: putamen, dACC: dorsal ACC, Pal: pallidum, vACC: ventral ACC, ac: anterior commissure, GPe: globus pallidus external segment, VP: ventral pallidum. Text indicates the plane's anterior-posterior position relative to the midline anterior commissure. Scale bar: 2 mm. **b** Reconstruction of recording sites. Circles indicate locations of neurons that responded to uncertainty with significant excitation (black) or inhibition (white). Structures are shown in the coronal plane. Neurons are projected onto the nearest shown plane; text indicates the range of neuron locations. **c** Example neurons in each area. The neurons have excitatory or inhibitory ramping activity up to the time of uncertain reward: strongest for highly uncertain rewards (blue, 50%), strong for intermediately uncertain rewards (turquoise colors, 25 and 75%), and weak for certain outcomes (black, 100%; gray, 0%). Top: spike times (dots) on each trial (rows). Bottom: smoothed firing rates. **d** Population average normalized activity of uncertainty coding neurons in each area. Shaded areas are ±1 SE. Arrow and text indicate latency of uncertainty coding. Gray area below x-axis is the pre-outcome analysis window. **e** Population average pre-outcome normalized activity is well fitted with a second-order polynomial function of reward probability (gray lines; shaded areas are ± 1 bootstrap SE) showing the inverted-U relationship with reward probability expected for uncertainty coding. **f** Graded coding of reward uncertainty. Histograms show each neuron's difference in normalized activity between CSs with high (50%) vs. intermediate (25, 75%) uncertainty. Colored neurons have significantly differential activity: more normalized activity for high uncertainty (blue) or intermediate uncertainty (turquoise). *** indicates more neurons than expected by chance ($p \le 0.001$, binomial test). Arrows indicate means of all neurons (open arrow) and all neurons with significant differential activity (filled arrows); **, *** indicate the significance of their difference from zero, $p < 0.01$, 0.001 (signed-rank tests)

uncertainty with ramping anticipatory activity. This raises a key question: what event is the network anticipating? Most crucially, does the network anticipate the moment of receiving an uncertain outcome per se, or the moment of receiving information to resolve the uncertainty?

To answer this question, we designed a task to separate the time of receiving information from the time of receiving the outcome (information task, Fig. 2a). On each trial, the monkey was shown a fractal CS that indicated that a reward would be delivered in 3 s with 100%, 50%, or 0% probability. There were two types of CSs. Information-predictive CSs (Info CSs) were followed after 1 s by an informative visual cue whose color indicated the upcoming outcome (e.g., orange → reward, gray → no reward). Noninformation-predictive CSs (Noinfo CSs) were followed by a noninformative cue whose color was randomized on each trial and hence did not indicate the upcoming outcome. Note that in this terminology the terms Info CS and Noinfo CS refer to whether the CS is followed by an informative cue (not to whether the CS itself conveys information about reward). Importantly, there was no way for animals to use the information to control or influence the outcome. Thus, neurons that anticipate the moment of receiving information to resolve uncertainty (Fig. 2b, Hypothesis 1) should be activated at distinct times on Info and Noinfo CS trials. On Info CS trials they should activate in anticipation of receiving information from the informative cue. On Noinfo CS trials, they should activate in anticipation of outcome delivery, because that is when the animal is first informed of the outcome (by receiving either juice or no juice). On the other hand, neurons that simply anticipate uncertain outcomes (Fig. 2b, Hypothesis 2) should respond identically

during the Info and Noinfo CSs because both types of CSs are associated with identical future reward outcomes (the same reward probability, amount, and timing). They should only differentiate between Info and Noinfo trials in anticipation of the outcome, when outcomes are certain on Info trials but uncertain on Noinfo trials (Fig. 2b, right, blue arrow).

Indeed, the cortico-BG network contained a substantial population of neurons that anticipated the moment of gaining information to resolve reward uncertainty. For example, the icbDS neuron in Fig. 2c closely resembled the hypothetical information-anticipatory signal (compare with Fig. 2b). This neuron was activated in advance of receiving information, both on Info CS trials during the CS period in anticipation of viewing an informative cue (top, red) and on Noinfo CS trials during the cue period in anticipation of reward delivery (bottom, blue). These activations were highly similar even though the information on these two trial types was conveyed through different modalities (visual cue vs. juice delivery). Importantly, this neuron had much lower activity during the same time periods when information was not expected (e.g., during the Noinfo CS in advance of noninformative cues, and during the informative cues in advance of the already-known outcome). The Pal neuron in Fig. 2d had a similar response, except with inhibitions rather than excitations. As in the original uncertainty task, these responses were not simply anticipating juice reward or encoding the value of the stimulus, because they were stronger when reward was uncertain than certain (50% > 100% reward).

We recorded from 154 uncertainty coding neurons using information tasks (ACC $n = 63$, icbDS $n = 24$, Pal $n = 67$; Methods). We defined a neuron's uncertainty signal as its ROC

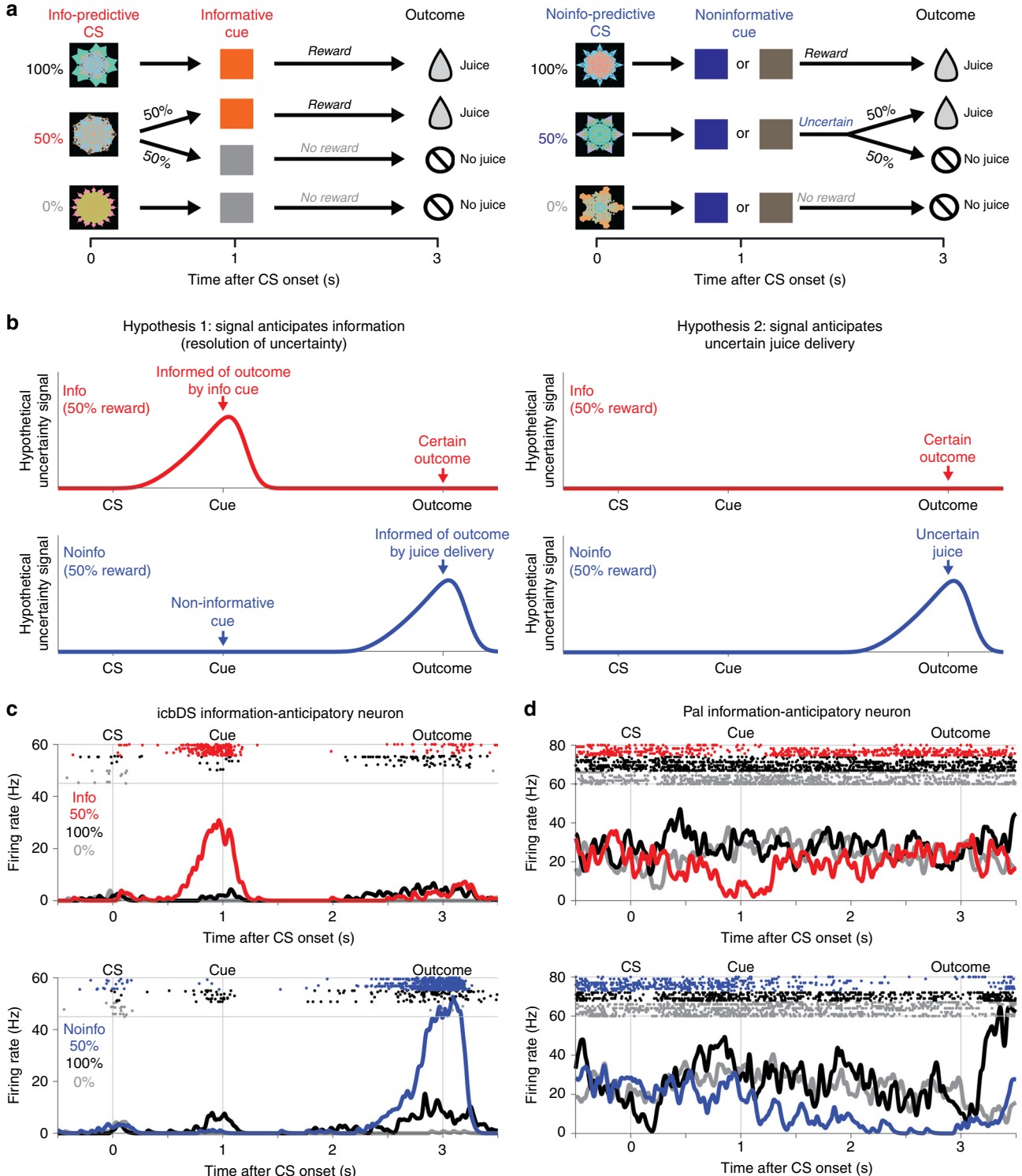

**Fig. 2** Neuronal activity anticipating the moment of gaining information to resolve reward uncertainty. **a** Information task. On Info CS trials, CSs predict 100, 50, or 0% reward, and are followed by an informative cue that indicates the outcome with certainty. On Noinfo CS trials, analogous CSs are followed by one of two noninformative cues that are randomized, and hence leave the outcome uncertain until it is delivered at the end of the trial. **b** Testing two hypotheses about the origin of uncertainty-related activity. In Hypothesis 1 (left), activity on 50% reward trials anticipates the moment of gaining information about the uncertain outcome, and hence anticipates both informative cues (top, red, Info CS) and uncertain outcomes (bottom, blue, Noinfo CS). In Hypothesis 2 (right), activity simply anticipates uncertain outcome delivery, and hence has no differential activity during the CS period because it only anticipates uncertain outcomes (bottom, blue, Noinfo CS). **c** An icbDS neuron with information-anticipatory activity. This neuron has strong ramping activation on 50% reward trials that anticipates informative cues (top, red, Info CS) and uncertain outcomes (bottom, blue, Noinfo CS). Its activity is greatly reduced or absent when the outcome is certain (100% reward, black; 0% reward, gray). **d** An example Pal neuron with information-anticipatory activity. This neuron has a similar response pattern but with ramping inhibition rather than excitation

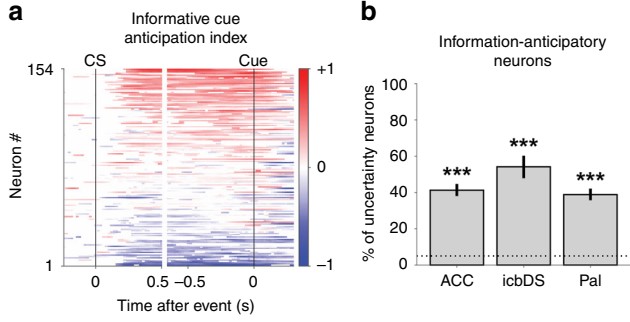

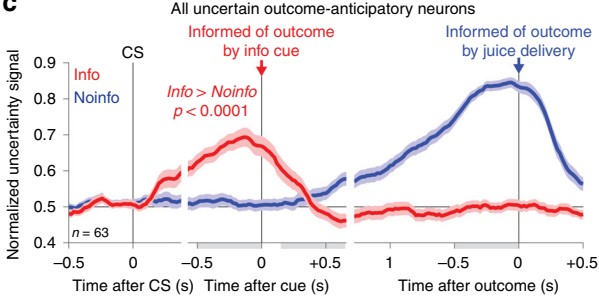

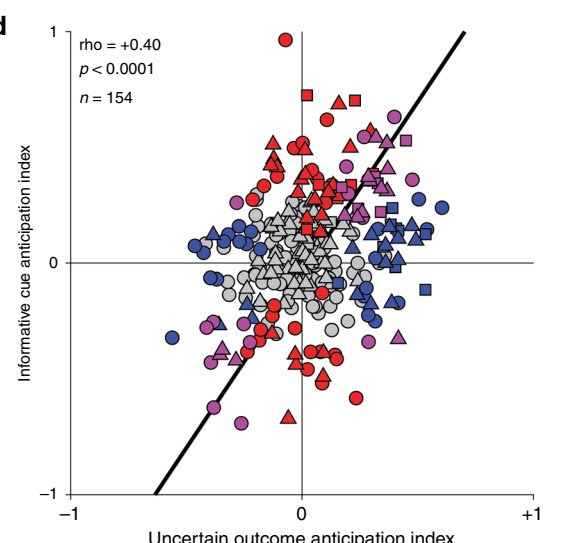

**Fig. 3** Prevalence of information-anticipatory activity in the cortico-basal ganglia network. **a** Uncertainty-related activity emerges preferentially during the Info CS in anticipation of informative cues. Each row is a neuron, and colors indicate time points with a significant Informative Cue Anticipation Index (red: more positive uncertainty signal for Info, top; blue: more negative uncertainty signal for Info, bottom; color bar indicates scale). **b** Information-anticipatory activity was significantly present in many uncertainty-related neurons in each area (ACC, icbDS, and Pal: $n = 26/63$, $13/24$, and $26/67$). *** indicates significantly more neurons than expected by chance, ($p < 0.001$, binomial test). **c** Information-anticipatory activity was prevalent in neurons anticipating uncertain outcomes. Shown are the population average uncertainty signals on Info CS (red) and Noinfo CS (blue) trials from the subset of neurons with a significant Uncertain Outcome Anticipation Index (blue, index measured only using Noinfo CS trials). The population has clear activity on Info CS trials in anticipation of viewing informative cues (red, text indicates $p$-value, rank-sum test). Error bars are ±1 SE. Gray-shaded areas are the time windows for calculating the indexes of cue anticipation (pre-cue window) and outcome anticipation (post-cue, pre-outcome windows). **d** Correlated anticipation of the two reward-informative task events. Many neurons have significant Informative Cue Anticipation Indexes (red, $y$-axis, $p < 0.05$, permutation test), Uncertain Outcome Anticipation Indexes (blue, $x$-axis), or both (purple). The two indexes are highly correlated; text indicates rank correlation and its $p$-value. ACC, icbDS, and Pal neurons are indicated by triangles, squares, and circles. Black line is a linear fit with type 2 regression

Anticipation Index as the change in a neuron's uncertainty signal from the beginning to the end of the cue period on Noinfo trials. This index was significant in substantial number of single neurons in all three areas (37%, 63%, and 34% of neurons in ACC, icbDS, and Pal; more neurons than expected by chance in all areas, binomial tests, all $p < 0.001$) and tended to be most prevalent in icbDS (higher fraction of significant neurons than ACC or Pal, $p = 0.0586$ vs. ACC, $p = 0.0313$ vs. Pal, permutation tests). There was a strong correlation between the two neural anticipation indexes (rank correlation = 0.40, $p < 0.001$, Fig. 3d). That is, many neurons were activated in anticipation of receiving information from both cues and outcomes, while other neurons were inhibited for both. Thus, when examining neurons whose uncertainty signals on Noinfo trials significantly anticipated the time of the outcome, the average time course of their uncertainty signals on Info trials bore a strong resemblance to a hypothetical information-anticipatory signal (Fig. 3c, compare with Fig. 2b). This was significantly different from hypothetical encoding of the current level of uncertainty or anticipation of uncertain outcome delivery (Supplementary Fig. 3). Similar results were found in all areas (Supplementary Fig. 4).

**Monkeys anticipate information to resolve uncertainty.** Given the network's information-predictive activity, we next asked whether information predictions evoke information-seeking behavior in monkeys. Monkeys, like humans, scan uncertain environments for information with their eyes. When information is available and can be obtained with a saccade, monkeys saccade with shorter latencies to view informative than non-informative cues[6,7]. However, it is unknown how monkeys behave at the time this network ramps up its activity: when information is not yet available, and monkeys can freely gaze in anticipation of the moment it will arrive. We hypothesized that monkeys anticipate information by directing their gaze to objects in their environment associated with the uncertainty to be resolved. Consistent with previous work, we found that monkeys anticipated juice rewards by licking (Supplementary Fig. 4), and their gaze was attracted to visual objects based on their juice reward value

area for using its firing rate to distinguish trials with uncertain rewards from certain rewards. Crucially, we classified neurons as uncertainty coding solely based on whether they had a significant uncertainty signal during a 0.5s time window before outcome delivery on Noinfo trials ($p < 0.05$, rank-sum test). We then independently asked whether the same neurons also had uncertainty signals before cue onset and on Info trials. Specifically, we calculated an Informative Cue Anticipation Index defined as the difference between the uncertainty signal on Info and Noinfo CS trials. In many neurons, this index became different from zero shortly after CS onset and built up to the time of the cue, in either an excitatory or inhibitory manner (Fig. 3a). We defined a cell as information-anticipatory if the index was significantly different from zero during the 0.5 s immediately before cue onset. Information-anticipatory neurons were highly prevalent in all three areas of the network (Fig. 3b).

Importantly, as expected for an information-anticipatory signal, the same neurons that anticipated informative cues on Info trials also commonly anticipated the time of uncertain outcomes on Noinfo trials, and did so in similar manners. To quantify this, we defined an analogous Uncertain Outcome

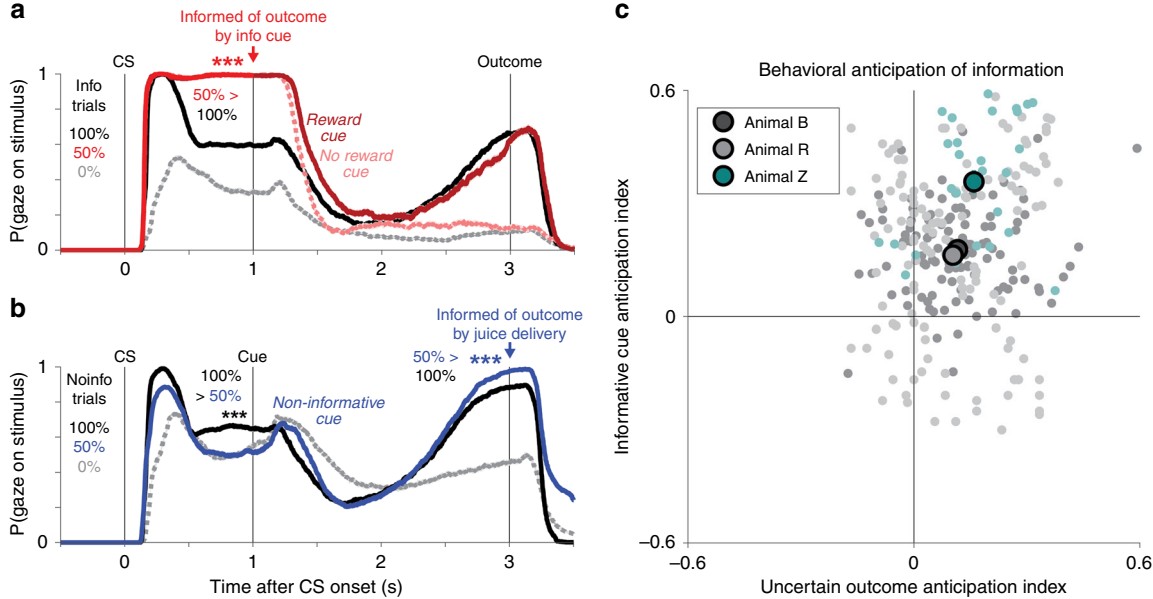

**Fig. 4** Monkeys anticipate the moment of gaining information by gazing at objects associated with the uncertain outcome. **a** Monkeys' gaze on Info trials is attracted to CSs in anticipation of receiving informative cues about uncertain rewards. Lines indicate the probability at each millisecond that the animal gazes at the stimulus. Shaded areas indicate ±1 SE (most too small to see). Monkeys gazed more at the 100% CS (black) than the 0% CS (gray), but gazed most of all at the uncertain 50% CS (red) which could be followed by informative cues indicating either reward (dark red) or no reward (pink). Gaze at reward cues then ramped up to the time of reward delivery, while gaze at no-reward cues was minimal. The data are from all $n = 68$ sessions of the version of the information task with the displayed task timing parameters. *** indicates $p < 0.001$ (signed-rank test) of the difference indicated by the text at the moment of a task event. **b** Gaze on Noinfo trials is attracted to objects in anticipation of outcome delivery. Same format as (**a**). During the CS period, monkeys had roughly value-based gaze behavior (100% > 50% > 0%), but during the cue period their gaze on 50% reward trials ramped up in anticipation of the uncertain outcome until it became greater than all other conditions (50% > 100% > 0%). **c** Uncertainty-related gaze behavior in each animal significantly anticipated informative cues (y-axis) and uncertain outcomes (x-axis). The data are from $n = 112, 151, 29$ sessions from animals A, R, Z. Same format as Fig. 3d, but analyzing gaze instead of neural activity. Colors indicate animals; light dots are single sessions; dark circles are means in each animal, error bars are ±1 SE (most too small to see)

(Fig. 4a, b, 100% CS > 0% CS, informative reward cue > no-reward cue).

Strikingly, however, monkeys' gaze was even more strongly attracted to objects based on their uncertainty, especially in the moments before receiving information to resolve that uncertainty. On Info trials, monkeys could anticipate receiving information during the Info 50% reward CS as they awaited the informative cue (Fig. 4a, red arrow). Indeed, monkeys were substantially more likely to gaze at the Info 50% reward CS than any other CS at the moment the cue was about to appear (signed-rank tests, all $p < 0.001$). Importantly, this attraction was specifically related to anticipating information, not reward value or uncertainty per se. Monkeys gazed at the Info 50% reward CS far more than the Noinfo 50% reward CS, which was associated with exactly the same reward value and uncertainty but was not followed by information (Fig. 3b, signed-rank test, $p < 0.001$), and far more than the 100% reward CSs, which were associated with double the reward value but no uncertainty (Fig. 4a, b; signed-rank test, $p < 0.001$). Furthermore, this avid gaze at the Info 50% reward CS occurred despite near-zero licking, indicating that animals were anticipating the delivery of information, not juice (Supplementary Fig. 4).

Similarly, on Noinfo trials, monkeys could anticipate receiving information at the end of the cue period as they awaited reward delivery or omission (Fig. 4b, blue arrow). Indeed, at that time monkeys gazed more at the cue during Noinfo 50% reward trials than all other conditions, even 100% reward trials that had double the reward value (signed-rank tests, all $p < 0.001$). This is remarkable because the information was delivered through a nonvisual modality (juice or no juice) and the 100%, 50%, and 0%

Noinfo trials all used exactly the same set of visual cue stimuli. Even so, monkeys gazed at those cues most avidly on Noinfo 50% reward trials in the moments before uncertainty was going to be resolved. Thus, when we analyzed monkeys' gaze in the same way, we analyzed neural spiking activity, we found that all monkeys had information-anticipatory behavior indicated by significantly positive Info Cue Anticipation and Uncertain Outcome Anticipation indexes (Fig. 4c, all $p < 0.001$, signed-rank test; Supplementary Fig. 5).

**Network activity predicts information-anticipatory behavior.** Further investigation revealed that the monkeys' information-anticipatory gaze was linked to moment-to-moment variability in the network's information-anticipatory signals. Examining time windows immediately before receipt of information, we found that neural information signals were present even at moments when the monkey's gaze was away from the visual stimulus, but were significantly stronger during matched time points from other trials when the monkey's gaze was on that visual stimulus (Fig. 5a).

Importantly, neural activity was not simply enhanced in a generalized manner whenever gaze was on any stimulus, nor was the enhancement the result of encoding simple visuomotor variables, such as gaze position or saccade direction (Supplementary Fig. 6). Instead, the enhancement took the form of a gain-like increase in the strength of neural information signals: activity was primarily enhanced when gazing at stimuli associated with uncertainty and its resolution, and was less affected when gazing at stimuli associated with certain outcomes (Fig. 5a; significantly

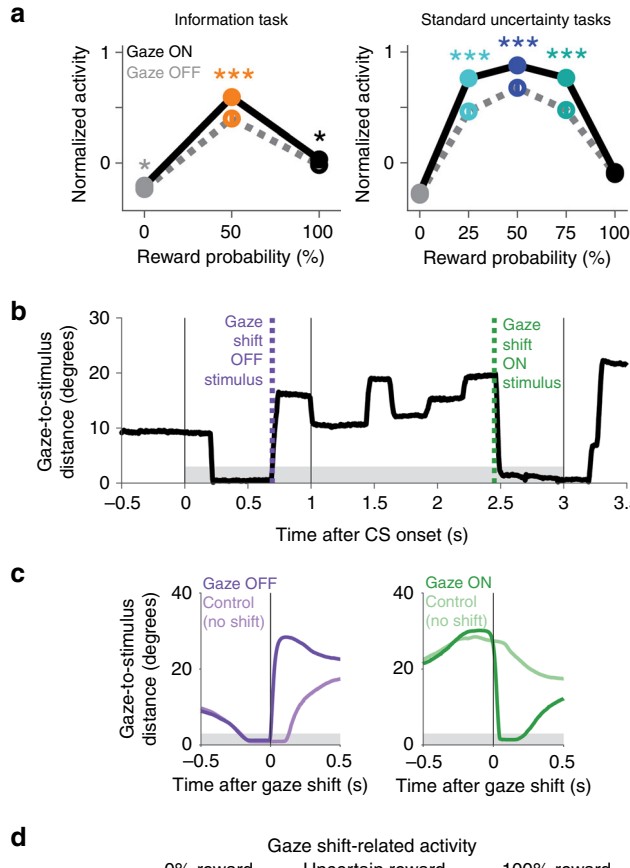

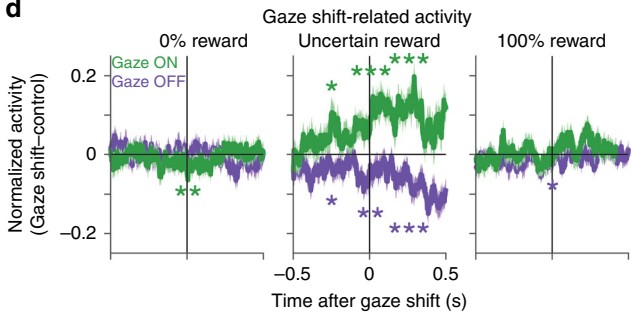

**Fig. 5** Fluctuations in cortico-BG network signals predict immediate gaze shifts toward or away from objects associated with resolving uncertainty. **a** Mean normalized activity before information is enhanced when gaze is on the stimulus (dark dots/lines) compared with off the stimulus (light dots/lines), especially for intermediate reward probabilities when reward is uncertain. Data are from information tasks (left) and standard uncertainty tasks (right). This analysis was performed on all neurons where there was trial-to-trial variance in gaze behavior before receipt of information for all of the CSs (information tasks $n = 117$: ACC $n = 62$, icbDS $n = 24$, Pal = 31; standard uncertainty tasks $n = 130$: ACC $n = 53$, icbDS $n = 53$, Pal = 24). Error bars are ±1 SE; *, *** indicate $p < 0.05$, 0.001 (signed-rank test). **b** Time course of gaze distance to the center of the stimulus (black) on an example trial with the Noinfo 50% reward CS. Gray area indicates the window for gaze to be on the stimulus. Green and purple lines indicate gaze shifts on and off the stimulus. **c** Mean time course of gaze distance to the stimulus aligned on gaze shifts off and on the stimulus (purple, green) and on control non-shift events (light colors), excluding blinks. Both gaze shift and control events have similar gaze trajectories until the moment of the gaze shift. **d** Mean time course of gaze shift-related normalized activity aligned on gaze shifts on the stimulus (green) and off the stimulus (purple). This analysis was performed on all neurons where there was at least one gaze shift on the stimulus and one gaze shift off the stimulus, as well as their matched control non-shift events, in each of three conditions: 0% reward (left), uncertain reward (middle), and 100% reward (right) ($n = 300$: ACC $n = 156$, icbDS $n = 78$, Pal $n = 66$). Shaded areas are ± 1 SE. *, **, *** indicate $p < 0.05$, 0.01, 0.001 (signed-rank test) before, during, and after the gaze shift. Uncertainty-related activity was enhanced before gaze shifts on the stimulus and reduced before gaze shifts off the stimulus, but primarily when animals were anticipating information about uncertain reward

initial gaze distance to the CS). The gaze shift trials and matched control trials had similar gaze trajectories up until the moment of the gaze shift (Fig. 5c). Strikingly, however, neural activity was significantly altered long before the gaze shift, with significant enhancement of activity before gaze shifts onto the stimulus and significant suppression of activity before gaze shifts off of the stimulus (Fig. 5d). This gaze-shift-related activity predominantly occurred when animals were anticipating information about an uncertain reward outcome; it was greatly reduced when reward was certain to be delivered or omitted (Fig. 5d). Also, this activity grew stronger in successive time windows over the course of the gaze shift (Fig. 5d; greater difference in activity between gaze shift on vs. off in each analysis time window compared with the previous window, $p < 0.05$, signed-rank test). Similar tendencies were present for all three regions and for all CS locations and gaze shift directions (Supplementary Fig. 6).

To quantify each neuron's modulation related to gaze and its time course relative to gaze shifts, we fit each neuron's activity with a model of its responses to all CS and cue conditions (see the Methods section). One set of parameters modeled the effects of gaze-related modulation (Fig. 6a, top): a multiplicative change in response gain (e.g., enhanced information-related signals, similar to Fig. 5a), an additive increase in firing rate (e.g., sensory or motor-related effects that have no effect on information signals), or a combination of both. A second set of parameters modeled the time course over which these modulations occurred relative to gaze shifts (Fig. 6a, bottom): neurons could modulate their activity either before or after gaze shifts, and their modulation could come online either rapidly or gradually (similar to Fig. 5d). The model accurately recovered the true effects and time courses of gaze-modulation in simulated data sets (Supplementary Fig 8).

The model fits indicated significant gaze-related modulation in 31% of neurons (permutation tests, $p < 0.05$; all areas above

greater effect of gaze on normalized activity on uncertain trials than certain trials, information task: $p = 0.0009$, standard uncertainty tasks: $p < 0.0001$, signed-rank tests). Thus, while information-anticipatory signals were present even after controlling for gaze state (Supplementary Fig. 7), they were particularly enhanced during gaze at visual stimuli associated with obtaining information.

These results suggested that the network's information signals are well suited to motivate information-seeking gaze shifts. We next set out to test this by investigating the link between neurons and behavior and through direct pharmacological disruption.

First, we asked whether neural information signals strengthened before gaze shifts, consistent with a causal role in motivating information seeking, or after gaze shifts, potentially reflecting a sensory response to the stimulus being brought closer to the fovea[29]. To test this, we pooled data from all uncertainty-related neurons recorded in all three areas of the network in all tasks ($n = 222, 127, 129$, in ACC, icbDS, and Pal; Supplementary Figs. 5–6; Supplementary Table 1). We aligned neural activity at the time of each gaze shift onto or off of the visual stimulus (Fig. 5b) and compared it with activity from matched control trials which had no gaze shift at that time but were matched in all other respects (i.e., same neuron, same CS, same cue, similar

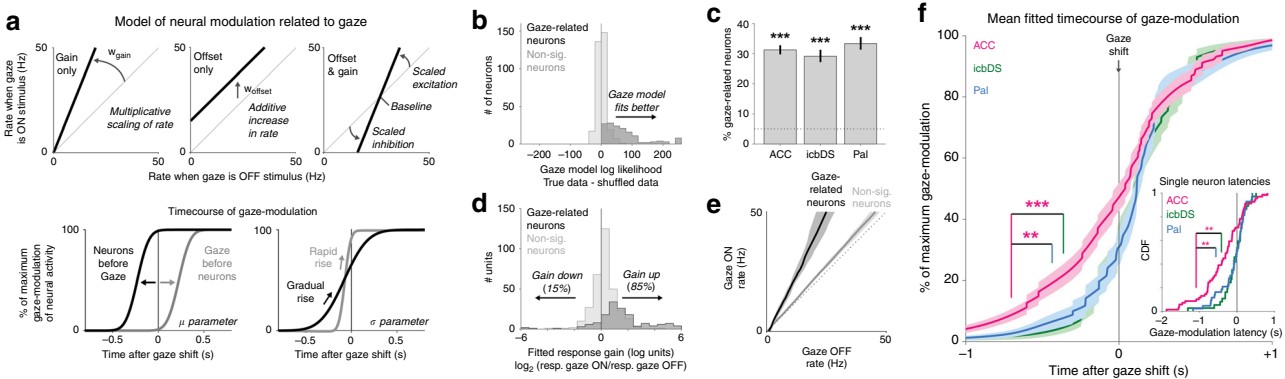

**Fig. 6 Prevalence and dynamics of gaze shift-related activity in the cortico-basal ganglia network. a** Model of gaze-related modulation in single neurons. Top: parameters fitting the effect of gaze on neural activity as multiplicative scaling (left), additive increase or decrease in firing rate (middle), or both (right). Bottom: parameters fitting the time course of gaze-modulation by setting the temporal offset between neural modulation and gaze shifts (left, neurons first vs. gaze first) and the rate that modulation builds up over time (right, gradual vs. rapid). **b** Model fits for each neuron evaluated by difference in log likelihood when fitting the real data vs. shuffled data. 31% of neurons were significantly modulated in relation to gaze (dark histogram, $p < 0.05$, permutation test); remaining neurons were not (light histogram). **c** Percentage of significantly gaze-related neurons (ACC $n = 69/221$, icbDS $n = 37/127$, Pal $n = 43/129$). Error bars are ±1 SE. *** indicates $p < 0.001$ (binomial test). **d** Fitted gaze-related response gains in each neuron in $\log_2$ units (0 = no gain change; −1 = halved response; +1 = doubled response). $n = 2$ outliers fitted with complete cessation of activity during gaze are plotted at −6. More neurons had gain increases than decreases, especially in gaze-related neurons (dark histogram, 127 increases vs. 22 decreases). **e** Fitted relationship between gaze state and firing rate in gaze-related neurons (black) and nonsignificant neurons (gray) resembles increased response gain rather than an additive offset. Lines are medians, shaded areas are bootstrap 95% confidence intervals. **f** Mean fitted time course of gaze-modulation around gaze shifts for neurons in each area with significant gaze-modulation (ACC, magenta; icbDS, green; Pal, blue). Asterisks indicate significance. The latency of gaze-modulation, defined at the time of 10% of maximal modulation, is significantly earlier in ACC than icbDS and Pal (gaze-modulated neurons, $p = 0.001$ and 0.003; all neurons, $p = 0.001$ and 0.003; permutation tests). Inset: cumulative distribution of gaze-modulation latencies for individual gaze-modulated neurons. Median latency is earlier in ACC than icbDS and Pal (gaze-modulated neurons, $p = 0.002$ and $p = 0.006$; all neurons, $p < 0.001$ and $p = 0.035$; rank-sum tests)

chance, binomial tests, $p < 0.05$, Fig. 6b, c). Gaze-modulation was best fit as including multiplicative changes in response gain rather than simply additive changes in firing rate, predominantly increases in gain rather than decreases (Fig. 6d). Thus, neural response strength increased during gaze at the stimulus, by a median of 33% across all neurons and 133% in significantly gaze-related neurons (Fig. 6d, e). These gain changes were present in all animals and tasks in which gaze was attracted by resolution of uncertainty (Supplementary Fig. 5). The mean time course of gaze-modulation in these neurons began long before the gaze shift in all areas, reaching 30–50% of maximal modulation before the eye movement occurred (Fig. 6f). Gaze-modulation began significantly earlier in ACC than icbDS and Pal. This was the case when comparing gaze-modulation latencies based on each area's mean time course of modulation (Fig. 6f, both $p < 0.01$, permutation tests) or latencies of modulation in individual neurons (Fig. 6f inset, both $p < 0.01$, rank-sum tests). Thus, while uncertainty signals emerged earliest in Pal (Fig. 1), fluctuations in ACC were first linked to future behavior.

Thus, the cortico-BG network has spontaneous fluctuations in its information-anticipatory signals. These changes start in ACC, continue in BG, and are followed by gaze shifts toward or away from the object associated with resolution of uncertainty.

**Perturbations of network activity impair information seeking.** Given these findings, we hypothesized that information-predictive neurons in the basal ganglia causally motivate gaze shifts to gain information. If so, temporarily inactivating the basal ganglia subregions that contain these neurons should impair the motivation to seek information. We therefore trained monkeys on a task in which gaze shifts were required to gain information (gaze-contingent information task, Fig. 7a)[6,7]. Monkeys fixated a spot of light and continued to fixate during a delay period while a

50% reward CS was presented on either the left or right. After the fixation point disappeared ("go" signal), the monkey was free to gaze in any manner they chose. On trials with an Info CS, gazing at the CS caused it to be immediately replaced by an informative cue indicating the trial's outcome. On trials with a Noinfo CS, gazing at the CS caused it to be replaced by a noninformative cue. Importantly, gazing at the CS allowed monkeys to gain information about the outcome but did not allow them to influence it in any way (e.g., the outcome always occurred a fixed time after the go signal regardless of whether and how they gazed at the CS). In addition, we encouraged animals to anticipate the moment information would be available by using a fixed duration between CS onset and the go signal. Indeed, animals had strongly anticipatory behavior, at times saccading to the CS at short latencies before they could have reacted to the go signal (Fig. 7b, shortest response times (RTs)). Monkeys were highly motivated to seek information, shown by faster RTs to gaze at Info CSs (Fig. 7b). We quantified this response bias favoring the Info CS with an Infobias Index (Fig. 7b) which was significantly positive in every session for every animal ($n = 43$ sessions, all $p < 0.05$, permutation tests). Furthermore, the network had similar information signals in the gaze-contingent task, indicating that its signals are present in this context where gaze shifts are required to gather information (Supplementary Fig. 9).

Based on the lateralized functions of basal ganglia circuitry[30,31], we predicted that unilateral inactivations would reduce information-seeking behavior directed toward objects in contralateral space (Supplementary Note 2). Indeed, unilateral injections of muscimol, a GABA$_A$ agonist, into either icbDS or Pal in the vicinity where information-anticipatory neurons were recorded caused the information-seeking response bias to be significantly reduced in contralateral space (Fig. 7c, d; icbDS, $n = 9$ sessions, $p = 0.028$; Pal, $n = 8$, $p = 0.031$; all inactivations, $n = 17$, $p = 0.0009$; permutation tests; Supplementary Figs. 2, 10; Supplementary Table 2). No

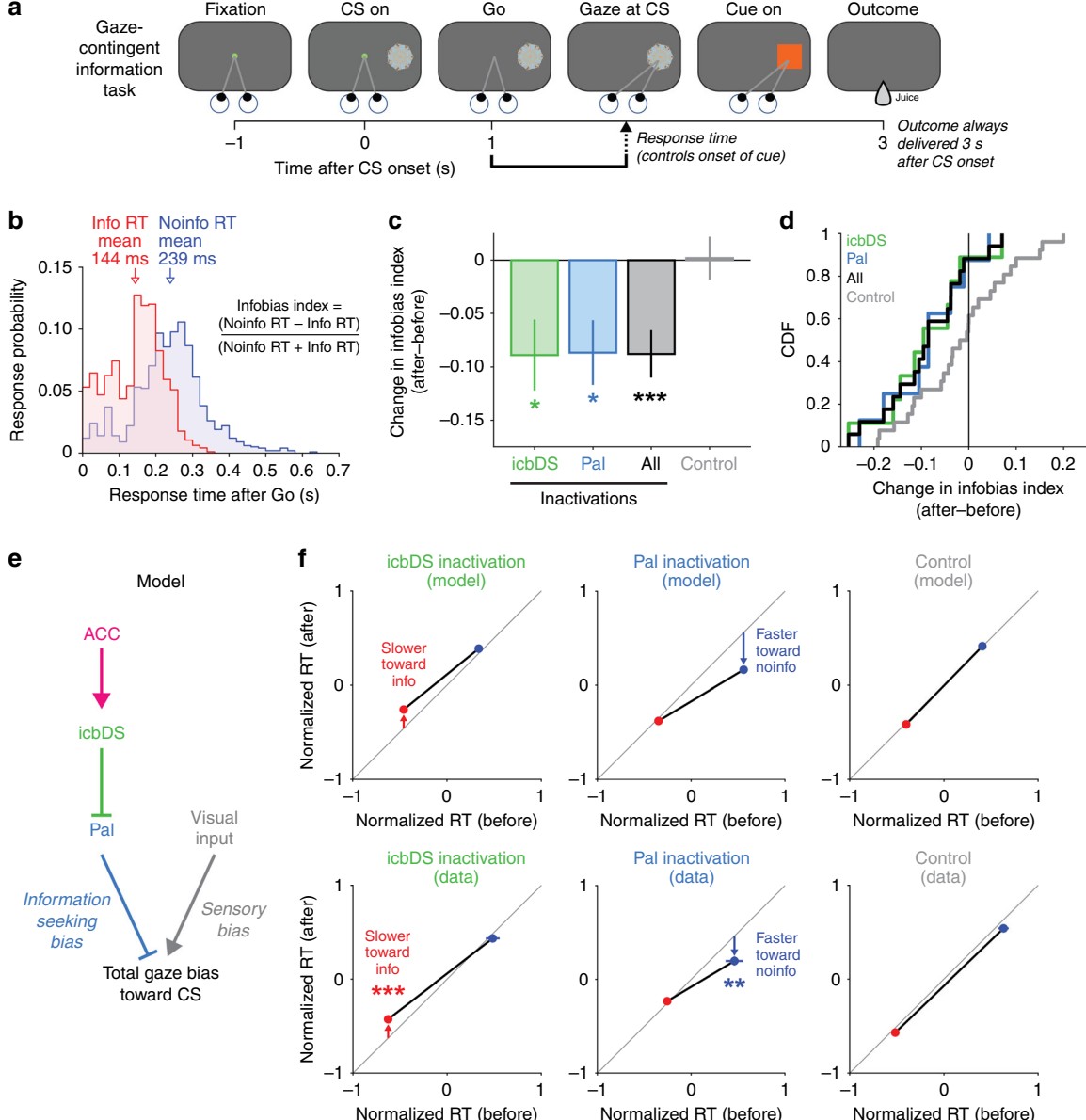

**Fig. 7** Perturbation of network activity impairs information seeking. **a** Gaze-contingent information task. Monkeys were shown a CS, waited for a go signal, and then were allowed to gaze at it. Gazing at an Info or Noinfo CS caused it to be replaced with an informative or noninformative cue. Regardless of gaze, the outcome was always delivered a fixed time after CS onset. **b** RT distribution for animal B. The animal had much faster RTs for Info trials (red) than Noinfo trials (blue). The animal often anticipated the time that information would become available, as indicated by the prevalence of anticipatory saccades especially on Info trials. This histogram includes all RTs collected when inactivations were not being performed ($n = 1966$), excluding four outliers from Noinfo trials (RTs = 0.741, 0.748, 1.326, 1.335 s). **c** Muscimol inactivation effect on RTs to contralateral CSs, quantified as the change in Infobias Index (after−before). There are significant reductions in Infobias Index for icbDS inactivations (green), Pal inactivations (blue), and all inactivations (black), but not control sessions (gray). *, **, *** indicate $p < 0.05$, 0.01, 0.001 (permutation tests). Error bars are ±1 SE. **d** Cumulative distributions showing each session's inactivation effect on the Infobias Index for contralateral CSs. Inactivation sessions consistently reduce the information bias; control sessions do not. **e** Schematic model of a mechanism by which the cortico-BG network could motivate information-seeking gaze shifts. **f** Model predictions: the two BG areas should influence information seeking in distinct manners, such that icbDS inactivation slows RTs to obtain information (left, Info CS, red), Pal inactivation speeds RTs that will not obtain information (middle, Noinfo CS, blue), and controls have no effect (right). **g** Inactivation results, quantified by comparing normalized RTs for the Info CS (red) and Noinfo CS (blue) before vs. after inactivation. icbDS inactivation slowed RTs to the Info CS (left); Pal inactivation speeded RTs to Noinfo CS (middle); control sessions had no significant effect on RTs to either CS (right). Error bars are ±1 SE. **, *** indicate $p < 0.01$, 0.001 (rank-sum tests).

significant change occurred in a control data set consisting of sham and saline injections (Fig. 7b, gray, $n = 26$ sessions, $p = 0.93$, permutation test). Thus, relative to control sessions, inactivations caused the information-seeking bias to be significantly reduced ($p = 0.005$, permutation test). In addition, inactivations had no significant effect on information seeking for ipsilateral CSs (all $p >$

0.4, permutation test; Supplementary Fig. 11). Thus, inactivations caused the information-seeking bias to become lateralized— significantly shifted away from the contralateral side ($p = 0.019$, permutation test; Supplementary Fig. 11).

We further investigated the mechanism by which icbDS and Pal activity promote information seeking. Our data indicate that

icbDS and Pal have reciprocal inhibitory connections and tend to encode information predictions in opposite manners, with icbDS neurons activated and Pal neurons commonly inhibited. We therefore hypothesized that icbDS and Pal activity have opposite influences on motivated gaze behavior, such that information-oriented gaze shifts are motivated by icbDS activity and suppressed by Pal activity. A simple network model implementing this hypothesis (Fig. 7e) reproduced the observed behavior on control sessions and predicted that inactivating icbDS and Pal should impair information seeking in distinct manners (Fig. 7f). The icbDS is primarily active during the Info CS, so inactivation should slow gaze shifts to the Info CS while leaving responses to the Noinfo CS relatively intact (Fig. 7f, left). Conversely, Pal is normally inhibited during the Info CS, so inactivation should leave responses to the Info CS relatively intact while speeding gaze shifts to the Noinfo CS (Fig. 7f, middle).

Both predictions were borne out in the data. icbDS inactivation slowed gaze shifts to the Info CS but did not significantly change RTs to the Noinfo CS (Fig. 7g, left, Info CS $p = 0.0003$, Noinfo CS $p = 0.64$, rank-sum tests; significantly different change in normalized RT for Info CSs vs. Noinfo CSs, indicated by a significant interaction term in a two-way ANOVA using the factors CS type (Info or Noinfo) and epoch (pre- or post-injection), $F_{1,1808} = 7.12$, $p = 0.008$; Methods). Conversely, Pal inactivation speeded gaze shifts to the Noinfo CS, but did not significantly change RTs to the Info CS (Fig. 7g, middle, Info CS $p = 0.17$, Noinfo CS $p = 0.001$, rank-sum tests; significantly different change in normalized RT for Info vs. Noinfo, two-way ANOVA, $F_{1,1955} = 7.89$, $p = 0.005$). Thus, a direct comparison between inactivations of the two areas revealed significantly different effects on behavior. icbDS inactivation slowed normalized RTs relative to Pal inactivation (permutation test, $p = 0.0043$). This occurred due to inactivations changing RTs in the predicted manner, and not in the orthogonal manner (significant effect of icbDS inactivation slowing saccades to the Info CS and Pal inactivation speeding saccades to the Noinfo CS, permutation test, $p < 0.0001$; no significant effect of icbDS slowing saccades to the Noinfo CS and Pal speeding saccades to the Info CS, $p = 0.3915$; Methods). These RT changes included anticipatory saccades, consistent with a disruption of information-anticipatory activity (Supplementary Fig. 12). Thus, icbDS activity motivated gaze shifts to gain information, while Pal activity suppressed motivation to gaze at objects that would not yield information.

Importantly, these inactivation effects on information seeking were not caused by generalized effects on overall motivation to perform the task, which could potentially be affected by inactivating adjacent circuits involved in primary reward-seeking behavior. Specifically, icbDS inactivations slowed RTs to the Info CS without reducing measures of general motivation, while Pal inactivations speeded RTs to the Noinfo CS without increasing measures of general motivation (Supplementary Fig. 11). If anything, Pal inactivations speeded RTs to the Noinfo CS in spite of a modest reduction in general motivation to perform the task, consistent with previous Pal inactivations[32–38].

## Discussion

Our work demonstrates a neural network that motivates actions to resolve uncertain situations by seeking knowledge about future rewards. Previous studies have identified cortical and basal ganglia networks that predict when future primary rewards are available and motivate behavior to seek those rewards[13,30,39,40]. Indeed, our monkeys had strong tendencies to gaze at visual stimuli based on their reward value. However, the subset of ACC–icbDS–Pal neurons we report here have relatively little

response to reward value, and hence their primary function is not likely to be control of such reward value-oriented behavior. Instead, they have a quite distinct function: they predict when information will become available to resolve reward uncertainty and motivate gaze behavior to obtain that information. This information-seeking gaze behavior can be even more potent than the attraction of gaze to primary reward: our monkeys gazed more avidly at the CS that provided informative cues than any other stimulus, even CSs and cues with double its expected reward value. Our data show that this information-seeking gaze behavior is tightly coupled to the cortico-BG network. Fluctuations in its information-anticipatory signals are followed by immediate gaze shifts toward or away from the information-related stimulus, and artificial perturbations of its activity interfere with information seeking in the manner predicted by its neural signals and connections.

Our data are crucial evidence for theories of reward learning, overt attention, and economic decision-making, which propose that objects and events are assigned salience both by neural systems that track primary reward value and its uncertainty[41–44], and a system that anticipates information to resolve uncertainty[2,6,45–47]. Furthermore, our data demonstrate a neural mechanism for information seeking to compete with primary reward to drive ongoing gaze behavior[7,9,10,48].

In fact, information seeking goes hand-in-hand with primary reward seeking in natural environments. Most experimental studies of reward seeking begin with the presentation of a cue stimulus (or context) that tells the subject what reward to predict and what actions are needed to obtain it. However, rewards in natural environments can be scarce and uncertain, and fully predictive reward cues rarely come for free or materialize from thin air. Organisms must first seek and obtain information about the rewards that are available in their environment; only then can they predict the value of those rewards and use their value to motivate reward-seeking behavior. In this sense, the cortico-BG network for information seeking may be critical to ensure that organisms seek out the reward-related cues in their environment that are necessary for the proper operation of the well-known networks that predict and seek primary rewards[13,30,39,40,49]. Indeed, information-related neurons in all three areas were intermixed with other neurons that encoded the reward value of stimuli and outcomes, as expected from previous studies of these areas[13,22,30,39,40,49–52].

This suggests that information- and primary reward-related neurons are well-positioned to support each other's computations. For instance, information-anticipatory activity in our tasks can be interpreted as ramping up to the expected time of a large reward-prediction error (evoked by being informed of a pleasing or disappointing outcome). Thus, while most uncertainty-related neurons in these areas do not encode reward-prediction errors themselves[22], their activity could prepare local reward-processing networks to handle upcoming reward-prediction error or surprise signals, processes in which ACC, dorsal striatum, and pallidum have been implicated[40,53–57]. Conversely, the network could learn its information-anticipatory activity by treating reward-prediction errors as a teaching signal, such that surprising outcomes evoke large prediction errors which lead to greater ramping on future trials, while correctly predicted outcomes evoke small prediction errors which lead to reduced ramping on future trials. This would help explain why icbDS ramping activity is strong during initial exposure to a novel, ambiguous situation, but rapidly diminishes if animals can learn to correctly predict its reward outcomes[21]. This learning process could be mediated by input from midbrain dopamine neurons, which encode phasic reward-prediction errors in response to informative feedback[13,58]. Other neuromodulators could serve this role as well. The network

receives acetylcholine from tonically active neurons in the striatum which respond during reward-prediction errors[59] and from the basal forebrain which contains neurons with activity related to surprise[60–62]; the network also receives norepinephrine from the locus coeruleus, which contains neurons with activity related to certain rewarding stimuli and actions[63–65] and is linked to pupil dilation[66] which can covary with surprise and uncertainty[67,68]. This raises the possibility that disruption of neuromodulators in the network could impair learning or performance of information-seeking behavior.

Importantly, monkeys expressed a strong preference for the information consistent with it having a high subjective value[9] even though the information did not have any objective value, in the sense that it did not help the monkeys take action to gain more juice reward (a quantity called the value of information or value of exploration in decision theory and reinforcement learning[69,70]). An important area of future research will be how these two factors combine to guide behavior and whether they are implemented by shared or distinct neural networks. Notably, our study identifies a network that motivates seeking of a specific type of information (i.e., the presence or absence of an upcoming reward). It remains unknown whether this network could also motivate seeking of other types of information, such as information about instrumental contingencies (i.e., what action will be required to gain the reward[71]). In natural environments, these types of information are likely to synergize with each other: if an agent has a subjective preference to resolve uncertainty in order to better predict future rewards, then the agent will likely be better at learning the objective value of resolving that uncertainty to better take actions and control rewards.

While information-anticipatory signals were present in all three areas of the cortico-BG network, each area also had distinct features suitable for unique contributions to information seeking. Notably, fluctuations in ACC information signals were the earliest predictor of future behavior. ACC information signals changed several hundred milliseconds before gaze shifts, while BG signals changed more proximally to behavior. This finding supports and extends theories that ACC is especially important for motivating behavioral shifts to explore available prospects and learn their reward value and other properties[72–74], tracking their level of uncertainty and how it evolves over time as beliefs are updated in response to surprising outcomes[12,73,75–79], and using this information to decide how to control future cognition and behavior[73,80]. In particular, while it is well acknowledged that the ACC needs to receive a broad array of reward- and uncertainty-related information to perform these functions[73,80], our data indicate that the ACC is not a mere passive recipient of this information. Rather, the ACC is tightly linked to the emergence of motivational drive to actively seek out that information from the environment. Information-anticipatory activity would be especially useful from the perspective of theories that the ACC regulates foraging[73], because knowing the properties of the potential sources of reward in the environment is a fundamental requirement for making efficient foraging decisions. It would also be useful from the perspective of theories that the ACC regulates cognitive control[80], because one of the most crucial times to bring cognitive resources online is in preparation for receiving a new piece of information, in order to process it quickly and prepare an effective response.

In addition, our findings indicate that information-seeking behavior is motivated by a BG circuit mechanism that is analogous but distinct from the BG circuits that motivate conventional reward-seeking behavior. There are two key parallels. First, behavior-related fluctuations in BG information signals follow fluctuations in the cortex and are proximal to behavioral gaze shifts. This is consistent with classic theories of cortico-BG circuits[25] and work on the cortex–striatum interactions[81–83], suggesting that cognitive and motivational signals can be computed in the cortex and then sent to BG whence they are processed and used to guide behavior. Second, the specific functions of each BG subregion in information seeking are consistent with classic BG circuit motifs (Fig. 7e): icbDS and Pal neurons commonly encode information signals with opposite signs and these areas have opposite causal influences on behavior, such that icbDS activity speeds gaze shifts to gain information while Pal activity slows gaze shifts that will not provide information. This resembles analogous findings for BG areas involved in primary reward seeking: antagonizing D1 receptors in visuomotor dorsal striatum slows gaze shifts to gain large juice rewards[84], while inactivation of Pal speeds gaze shifts to gain small juice rewards[35].

Importantly, however, the BG mechanisms underlying physical reward- and information-oriented behavior are at least partially distinct at the neuronal and behavioral levels: first, when animals avidly gazed at the Info CS in anticipation of viewing the informative cue, they had near-zero licking behavior indicating that they were not anticipating juice reward; second, the cortical and BG neurons we identified that are linked to information-anticipatory behavior primarily anticipated the moment of gaining information, not the moment of gaining juice reward; third, inactivation effects on information seeking could not be explained as a result of generalized effects on juice reward seeking. Thus, this cortico-BG network appears to be specially focused on online information-seeking behavior. This is in contrast to other BG circuits and interconnected areas involved in reward-prediction errors and reinforcement, which commonly encode information and primary reward in a common currency[6,11].

In addition, whereas classic theories of cortico-BG circuits classify Pal as an output structure[25,85], our data extend recent results[37,38,86] by showing that Pal in fact responds earliest to uncertainty-related events. This is consistent with theories that BG rapidly selects salient stimuli to be used to guide future behavior[87,88], perhaps based on input from areas specialized for rapid assessment of objects and their incentive properties, such as the amygdala[89] and brainstem[90]. Our data support a scenario, in which (A) Pal rapidly signals a rough assessment of reward uncertainty; (B) ACC and icbDS next signal the precise graded level of uncertainty; (C) the resulting representation of uncertainty in all three areas ramps up to the time of its resolution by information, and drives ongoing information-seeking behavior.

Given the link between the ACC–icbDS–Pal network and information-seeking behavior, variations in the network's activity could be responsible for the natural variations in information-seeking behavior that are commonly found across individuals[5,10,12] and tasks[3,91]. In the same vein, it is notable that ACC and BG are implicated as sites of dysfunction and targets for treatment in human disorders of motivated behavior (such as obsessive–compulsive disorder[92,93], Parkinson's disease[94], and drug addiction[95]) that are known to affect reward- and uncertainty-related behavior[96–101]. Our results raise the possibility that these disorders and treatments may also affect the motivation to seek information about future events. While this has been little studied, there is evidence that Parkinson's disease reduces the motivation to gather information needed for upcoming decisions[102] and impairs learning from early access to information about uncertain outcomes[103]. Taken together, our work provides a foundation for understanding the neural network mechanisms by which information is detected, predicted, and used to motivate behavior.

## Methods

**General procedures.** Four adult male rhesus monkeys (*Macaca mulatta*) were used for behavioral, recording, and inactivation experiments (Animals B, R, Z, and

W). All procedures conformed to the Guide for the Care and Use of Laboratory Animals, and were approved by the Washington University Institutional Animal Care and Use Committee. A plastic head holder and plastic recording chamber were fixed to the skull under general anesthesia and sterile surgical conditions. The chambers were tilted laterally by 35–40° and aimed at the anterior cingulate and the anterior regions of the basal ganglia. After the animals recovered from surgery, they participated in the experiments.

**Data acquisition**. While the animals participated in the behavioral tasks, we recorded single neurons in the anterior cingulate cortex (ACC), internal-capsule-bordering regions of the dorsal striatum (icbDS), and anterior pallidum, including the ventral pallidum and the anteriormost part of the globus pallidus external segment (Pal). Electrode trajectories were determined with a 1-mm spacing grid system and with the aid of MR images (3 T) obtained along the direction of the recording chamber. This MRI-based estimation of neuron recording locations was aided by custom-built software (PyElectrode[104]). In addition, in order to further verify the location of recording sites, after a subset of experiments the electrode was temporarily fixed in place at the recording site and the electrode tip's location in the target area was verified by MRI (Supplementary Fig. 2).

Single-unit recording was performed using glass-coated electrodes (Alpha Omega). The electrode was inserted through a stainless steel guide tube and advanced by an oil-driven micromanipulator (MO-97A, Narishige). Signal acquisition (including amplification and filtering) was performed using an Alpha Omega 44 kHz SNR system. Action potential waveforms were identified online by multiple time-amplitude windows with an additional template-matching algorithm (Alpha Omega). This recording was restricted to single neurons that were isolated online. A subset of Pal neurons ($n = 36$) were instead recorded using 16-channel V-probes (Plexon) and isolated offline using Offline Sorter (Plexon). Neuronal and behavioral analyses were conducted offline in Matlab (Mathworks, Natick, MA).

Eye position was obtained with an infrared video camera (Eyelink, SR Research). Behavioral events and visual stimuli were controlled by Matlab (Mathworks, Natick, MA) with Psychophysics Toolbox extensions. Juice, used as reward, was delivered with a solenoid delivery reward system (CRIST Instruments). We monitored the magnitude of anticipatory mouth movements using a strain gauge attached to the juice spout. To detect licks, the strain gauge signal was acquired at 1 kHz, converted to its absolute value, and smoothed with an eighth-order low-pass Butterworth filter with 10 Hz cutoff frequency. Licks were defined as moments during a trial when the strain gauge signal rose above a threshold. The lick threshold was set for each session based on the distribution of baseline signals in that session, as follows. The baseline signal was measured on each correct trial as the mean signal in a 400 ms window beginning at the start of each trial (before the onset of the central trial start cue). The lick threshold was then set equal to the mean of the baseline signals plus two times their SD.

**Behavioral tasks**. We analyzed data recorded from several behavioral tasks which can be grouped into two major categories: standard uncertainty tasks and information tasks.

The standard uncertainty tasks are described in detail in previous work[21–23] and described again here. They each used a distinct set of fractal visual CSs with different associated outcomes. However, they all shared the following general outline. Animals were presented with a small white circular trial start cue at the center of the screen. In some tasks, animals were required to fixate the trial start cue for a fixed duration (typically 0.5–1 s) for the trial to continue; if they failed to fulfill this requirement within a grace period (typically 5 s), the trial would be considered an error, they would receive a timeout, and the trial would repeat. In other tasks, animal were not required to fixate the trial start cue; it was simply shown for a fixed duration (typically 1 s). After the trial start period, the trial start cue disappeared, and a fractal visual conditioned stimulus (CS, 2° radius) appeared on the screen for a fixed duration (2.5 s). The CS was randomly positioned at one of three locations: the center of the screen, the left side of the screen, or the right side of the screen (at 10° or 12.5° eccentricity). In some sessions, only the left and right locations were used. Animals were not required to gaze at or interact with the CS in any way. At the end of the CS period, the CS disappeared, and simultaneously the trial's outcome was delivered. Finally, there was an intertrial interval during which the screen was blank (typically randomized between 1–8 s, with different durations for different animals and tasks). In some sessions, a small fraction of intertrial intervals included the unexpected presentation of salient events, which could be appetitive (juice), aversive (an airpuff, ~35 psi, delivered through a narrow tube placed ~6–8 cm from the face[22]), or audiovisual (an auditory tone sounding and the screen flashing white).

The standard uncertainty tasks primarily differed in their CSs, outcomes, and block structure:

Task A:[21,22] Trials were presented in two distinct blocks. In the Probability block, there were five CSs associated with 0, 25, 50, 75, and 100% probabilities of 0.25 mL of juice. In the Amount block, there were five CSs associated with 100% probability of 0, 0.0625, 0.125, 0.1875, and 0.25 mL of juice. Hence for each CS in the Probability block, there was a matched CS in the Amount block that was associated with an identical mean amount of juice, but for which the

outcome was certain rather than probabilistic. Each block consisted of 20 trials (four presentations of each of its five CSs, shuffled in a randomized order). The two blocks were presented repeatedly in an alternating manner, with each block continuing until its 20 trials were correctly completed and then immediately transitioning into the other block.

Task B:[21,23] Same as task A, except it used three Probability CSs (0, 50, 100%) and the three corresponding Amount CSs (0, 0.125, 0.25 mL), and each block consisted of six or nine trials (two or three presentations of each of its three CSs). In some sessions, blocks also included interleaved choice trials, in which two CSs were presented and animals were allowed to choose between them with a saccade; our analysis here is on non-choice trials.

Task C:[21,22] Trials were presented in two distinct blocks. In the Appetitive block, there were three CSs associated with 0, 50, and 100% probabilities of 0.4 mL of juice. In the Aversive block, there were three CSs associated with 0, 50, and 100% probabilities of airpuffs. Each block consisted of 12 trials (four presentations of each of its three CSs).

Task D:[22] There were nine CSs. Four CSs were associated with 25, 50, 75, and 100% probabilities of 0.4 mL of juice. Four other CSs were associated with 25, 50, 75, and 100% probabilities of airpuff. One final CS was associated with no outcome (i.e., 0% probability of both reward and airpuff). The CSs were presented in a pseudorandom order.

Task E: Three CSs were associated with 0, 50, and 100% probabilities of 0.25 mL of juice. The CSs were presented in a pseudorandom order.

The information tasks follow the design in Fig. 2a or are variants of this procedure. The task began with the appearance of a small circular trial start cue at the center of the screen, in which animals were required to fixate for a fixed duration (typically 0.5 or 1 s). The trial start cue then disappeared and was followed in succession by a CS (2° radius) that was displayed for a fixed duration (typically 1 s), which was replaced by a cue of the same width and height at the same location that was displayed for a fixed duration (typically 2 s). The cue then disappeared, and simultaneously the outcome was delivered. The trial then completed with a 1 s intertrial interval. The CSs were presented randomly on either the left or right side of the screen (10°). The CSs came in two types. The Info CSs predicted juice reward (typically 0.25 mL) with different probabilities (e.g., 0, 50, and 100%), and were followed by one of two informative cues whose color or texture indicated the trial's outcome. The Noinfo *CSs* also yielded juice reward with matched size and probability, but were followed by one of two noninformative cues whose colors or textures were randomized on each trial and hence did not convey any information about the trial's outcome. In some sessions, Noinfo CSs were followed by a single noninformative visual cue; there was no apparent difference in behavior or neural activity between sessions with one or two noninformative cues and hence their data was pooled. The CSs were presented in a pseudorandom order.

We collected data using the following information tasks:

Task IA: the task shown in Fig. 2a, used to record the majority of neurons. There are three Info CSs and three Noinfo CSs, respectively associated with 0, 50, and 100% probability of reward.

Task IB: the gaze-contingent information task, shown in Fig. 7a. This followed the same general procedure as task IA, but with a few modifications. The trial start cue remained visible for a fixed duration after CS onset during which animals were required to maintain fixation on the trial start cue (typically for 1 s in animal B, 0.25 s in animal R, and 0.5 or 1 s in animal Z). Fixation breaks were treated as errors: the screen went blank, there was a 1–2 s penalty delay period, and then the trial repeated from the beginning. After the fixation period, the trial start cue disappeared, and animals were free to move their eyes. The task then detected the first moment when animals gazed at the CS, defined as the eye position entering a square window centered on the CS (i.e., when horizontal and vertical eye positions were within 4° of the center of the CS). If animals gazed at an Info CS, it was immediately replaced with the appropriate informative cue; if they gazed at a Noinfo CS, it was immediately replaced with a noninformative cue; if they did not gaze at a CS, no cue was shown. Importantly, regardless of their gaze behavior, all stimuli disappeared and the outcome was delivered at the same, fixed time after CS onset on all trials in the session (typically 3 s). Thus, gazing at the CS gave animals access to the cues, but did not give them earlier access to the juice reward. In the version of this task used for neuronal recording, we used the same visual CSs as in Task IA. Tasks IA and IB were typically pseudo-randomly interleaved in a trial-by-trial manner. At the start of each trial, the current task was indicated to the animal by the color of the fixation point (white for IA, green for IB). In the version of this task used for inactivation experiments and controls, there were only two CSs —an Info CS and a Noinfo CS—that were both associated with 50% probability of 0.25 mL of juice reward. This was to minimize the possibility that gaze behavior to the CSs could be influenced by different reward expectations or reward-prediction errors induced by CS onset, by ensuring that that the probability, amount, and timing of juice reward was identical for all CSs on all trials.

Task IC: used to record a subset of neurons in animal B. Similar to task IA but instead of three types of CSs (0, 50, and 100% probability of reward) there were ten types of CSs, which were respectively associated with 25, 50, 75, or 100%

probabilities of reward, with the equivalent probabilities of punishment, with no outcome (i.e., 0% probability of either reward or punishment), or with 50% probability of either reward or punishment. Info and Noinfo trials were presented in separate blocks. There were also minor changes in task timing and visual stimuli: the CS and cue periods were 1 s and 2.25 s duration, cues were presented as colored rectangles inside or around the CS rather than as squares replacing the CS, and the CSs remained onscreen during the first 1.5 s of the cue period.

Task ID: used to record a subset of neurons in animal R. Similar to task IA, but with only three total CSs: Info 50% chance of 0.38 mL reward, Noinfo 50% chance of 0.38 mL reward, and a Certain CS which yielded a 100% chance of 0.19 mL reward. For this task, uncertainty signals were defined on Info trials as the ROC area comparing Info 50% CS trials to Certain CS trials, and on Noinfo trials as the ROC area comparing Noinfo 50% CS trials to Certain CS trials. There were also minor changes in task timing and visual stimuli: the CS and cue periods were 1.5 s and 1.5 s in duration, each individual CS was associated with two distinct cues, and the cues were square-shaped fractal stimuli rather than colored rectangles.

Information-related neural activity was typically similar in the standard and gaze-contingent versions of the information task (IA and IB), e.g., activity ramping up to the time the informative cue would become available and to the time a noninformed outcome would be delivered. Note, however, that the task design of the gaze-contingent task potentially induced a link between gaze and receipt of information: gaze behavior was not completely "free" because it was required if the animal wanted to produce the cue, and the cue appeared with variable timing depending on the animal's behavior. Therefore, to be conservative, data from the gaze-contingent task was excluded from all of our analysis of neural-behavioral links (Figs 5–6), and was only used for a subset of other analyses. First, we analyzed the data from tasks IA and IB separately to compare them to each other in our supplementary analysis, testing whether information signals are altered when access to information is explicitly contingent on gaze. Second, a neuron's uncertainty coding in the time window 0.5 s before outcome delivery was calculated by pooling data from tasks IA and IB, because by that time in the trial the task was no longer gaze-contingent because animals had already revealed the cue on the great majority of trials. All remaining analyses, including our main analysis of information signals (Figs. 2–3), were restricted to the data from task IA, except for a small number of neurons for which the data from task IA was not available because they were only recorded in task IB ($n = 4$ ACC neurons in monkey Z).

Analysis of neural activity in Fig. 1 and Supplementary S1 uses the data from all neurons recorded using standard uncertainty tasks in which CSs were associated with all five reward probabilities (tasks A and D). Analysis of neural activity during information tasks (Figs. 2, 3) uses the data from neurons recorded using an information task. Analyses of neural activity related to gaze behavior (Figs. 5, 6) pooled the data from all neurons recorded in all tasks that contained trials of the types specified in the analyses (e.g., Fig. 5a used neurons with the data from 0, 50, and 100% conditions on both Info and Noinfo trials), except for the four ACC neurons described above from animal Z recorded only in the gaze-contingent information task, and six Pal neurons from animal W recorded in task A for which the gaze measurements were excessively noisy due to an error in configuring the eye tracker. If a neuron was recorded in multiple tasks, its data were fitted separately for each task and then pooled over tasks. Specifically, for each neuron, the model's total log likelihood was calculated as the sum of the log likelihoods from the individual tasks, and the neuron's gaze-aligned activity (e.g., Fig. 5), fitted gaze-related gain change and fitted time course of gaze-modulation and latency of gaze-modulation (e.g., Fig. 6) were calculated for each task and then averaged over tasks.

**Muscimol injections.** On muscimol injection sessions, a 33-gauge cannula was inserted through a 23-gauge guide tube into a grid hole and to a depth previously identified to be in icbDS or Pal and to contain information-related neurons (see Supplementary Fig. 2 and Supplementary Table 2 for coordinates of all injection sites). The other end of the cannula was connected to a 10 -μL Hamilton syringe. Behavioral data from the gaze-contingent information task were collected in blocks of 70–150 correct trials. Before the injection, we collected a "pre-injection" behavioral data set from the animal performing the task, typically for one block (median: 96 correct trials, standard deviation: 24, range: 48–164). After recording the baseline data, we used a manual syringe pump (Stoelting) or automated syringe pump (Harvard Apparatus) to inject muscimol dissolved in saline. Muscimol concentrations were 8 mg/mL, injection rates were typically 0.1 μL/min (range: 0.09–0.2), and injection volumes are reported for each session in Supplementary Table 2. After each injection, we collected a "post-injection" behavioral data set (median: 303 correct trials, standard deviation: 157, range: 43–839). All pre- and post-injection blocks of the data were included in our analysis regardless of the animal's response times or other gaze behavior, as long as the animal remained engaged in the task (i.e., generally initiating trials quickly and performing them correctly). On two sessions, pre-injection data from the same day were not available, so to obtain a comparable baseline we used the first block of behavioral data collected from the same animal on days immediately before or after the session. On saline injection control sessions, the same procedure was followed, except that only the saline vehicle was injected (with the same volumes previously used for

muscimol injections). On sham control sessions, the same procedure was followed to mimic the procedure for injection experiments in every detail, except that no cannula was inserted and no injection was made. Specifically, we (1) set up the injection equipment above the animal, (2) mounted the microdrive, (3) closed the experimental booth and waited for the standard period of time that would be required to use the microdrive to advance the tip of the cannula to the target area, (4) started the behavioral task and ran it for the same duration and schedule as the "before" condition on injection days, (5) stopped the task, entered the booth, and turned on the motor of the injection device to mimic the sounds and duration of time spent performing an injection, (6) turned on the task again and ran it for the same duration and schedule as the "after" condition on injection days. The only difference was that in injection experiments, the cannula was loaded into the microdrive and advanced to the target area, while in sham experiments it was not. Thus, sham sessions act as a control for the possibility that the observed alternations in information seeking were due to a generalized effect of the experimental procedure.

**Data analysis.** All statistical tests were two-tailed unless otherwise noted. Neurons recorded in the standard uncertainty tasks were included in our data set if they showed significant responsiveness to uncertainty (activity on uncertain reward CS trials significantly different from both 0% reward CS trials and 100% reward CS trials, rank-sum tests, both $p < 0.05$ and both differences with the same sign). For this purpose, activity was measured in a broad time window encompassing the CS period in order to avoid making any assumptions about the time course of neural responses (0.1–2.5 s after CS onset). The neuron's sign of uncertainty coding was defined as $+1$ if its ROC area for discriminating between uncertain reward CS trials vs. the pooled data from 0 and 100% reward CS trials was $> 0.5$, and defined as $-1$ if its ROC area was $< 0.5$. Similarly, for the information task, to avoid making any assumptions about the nature of Info-, CS-, or cue-related activity, neurons were included in our data set if their firing rates in a time window 0.5 s before outcome delivery significantly discriminated between Noinfo certain vs. uncertain reward trials (ROC area $\neq 0.5$, $p < 0.05$, rank-sum test), and its sign of uncertainty coding was set based on this activity in the analogous manner.

Neural activity was converted to normalized activity as follows. Each neuron's spiking activity was smoothed with a causal exponential kernel (mean $= 30$ ms) and then z-scored and sign-normalized using the following procedure. The neuron's average activity time course aligned at CS onset was calculated for each condition (defined here as each combination of CS and cue). These average activity time courses from the different conditions were all concatenated into a single vector, and its mean and standard deviation were calculated. Henceforth, all future analyses converted that neuron's firing rates to normalized activity by (1) subtracting the mean of that vector, (2) dividing by the standard deviation of that vector, (3) multiplying by the neuron's sign of uncertainty coding. Thus, normalized activity of $+1$ in a given task condition means that the neuron's firing rate deviated away from its average firing rate in the same direction that it responded to uncertainty, by an amount equivalent to 1 SD of its overall distribution of average firing rates during the task.

Neural uncertainty signals were calculated in specific time windows (e.g., pre-cue, pre-outcome, etc.), as the ROC area for distinguishing activity on uncertain reward CS trials (25, 50, and 75%) from pooled data from 0 and 100% certain reward trials. In the information tasks, uncertainty signals were calculated separately for Info and Noinfo trials. To visualize their time courses, they were calculated on neural activity at millisecond resolution after activity was smoothed with a gaussian kernel (SD $= 50$ ms) and sign-normalized based on the neuron's sign of uncertainty coding on Noinfo trials in a 0.5 s pre-outcome window (Fig. 3). The Informative Cue Anticipation Index was defined as the difference between its uncertainty signal for Info and Noinfo trials in a 0.5 s pre-cue time window (or for Fig. 3a, visualized by calculating it separately at each time point). Hence the index was positive if a neuron had a higher uncertainty signal in anticipation of Info CSs, and negative if a neuron had a higher uncertainty signal in anticipation of Noinfo CSs. The Uncertain Outcome Anticipation Index was defined as the difference between its uncertainty signals computed on two different time windows on Noinfo trials: a 0.5 s pre-outcome window, and a 0.5 s post-cue window (0.15–0.65 s after cue onset). Hence the index was positive if a neuron's uncertainty signal grew more positive between the cue and outcome, and negative if it grew more negative between the cue and outcome. Neurons were classified as information-responsive if their Informative Cue Anticipation Index was significantly different from 0 ($p < 0.05$, permutation tests conducted by comparing the index calculated on the true data to the distribution of indexes calculated on 20,000 permuted data sets which shuffled the assignment of trials to Info and Noinfo conditions). Neurons were classified as having a significant Uncertain Outcome Anticipation Index using the analogous permutation test ($p < 0.05$, shuffling the assignment of the data to the post-cue and pre-outcome time windows). For analysis of information-oriented gaze behavior, the same two indexes were calculated for each neuron except that instead of using neural data they used the behavioral gaze data from the last millisecond in each time window (equal to 1 for milliseconds when the animal's gaze was classified as being in the stimulus window and 0 otherwise). Finally, to plot the time course of uncertainty signals from the population including neurons with different signs of uncertainty coding, the normalized uncertainty signal (Fig. 3c) was calculated for each neuron

using the equation:

$$U_N = 0.5 + (U - 0.5)S, \tag{1}$$

…where $U_N$ is the normalized uncertainty signal, $U$ is the uncertainty signal, and $S$ is the neuron's sign of uncertainty coding ($-1$ or $+1$). Thus, neurons with a positive sign of uncertainty coding (e.g., excited by uncertainty) had their uncertainty signals left intact, while neurons with a negative sign of uncertainty coding (e.g., inhibited by uncertainty) had their uncertainty signals flipped. This ensured that each neuron had a positive normalized uncertainty signal in the time window and task condition that was used to classify its sign of uncertainty coding.

**Analysis of latency of uncertainty coding.** In standard uncertainty tasks, each neuron's smoothed normalized activity aligned at CS onset was further smoothed with a 101 ms causal boxcar kernel and then tested at each millisecond starting 50 ms after CS onset for whether it met the following criteria: (1) highly significant ROC area for distinguishing pooled data from uncertain reward CSs from the certain 0% reward CS ($p < 0.005$), (2) highly significant ROC area for distinguishing pooled data from uncertain reward CSs from the certain 100% reward CS ($p < 0.005$); (3) both ROC areas have the same "sign" (i.e., both > 0.5 indicating activation by uncertainty or < 0.5 indicating inhibition by uncertainty). A neuron's uncertainty coding latency was defined as the first millisecond after which it met these criteria for at least 24 consecutive milliseconds. In the information tasks, latencies were calculated in this manner separately for Info trials and Noinfo trials, and the neuron's overall latency was set to be the shorter of the two. In information task ID, the first criterion was not applied because there was no task condition with a 0% chance of reward. See Supplementary Fig 1 for the latencies and full ROC time courses in all neurons with detected latencies. This method was chosen to produce latencies that resemble those seen in raw traces of neural activity, but the same key result (i.e., Pal having shorter latency than ACC and icbDS) was found with other latency detection methods (e.g., different smoothing methods, significance criteria, required number of consecutive time bins, etc). Each area's latency was defined as the shortest latency of its single neurons after excluding the shortest 1% of single-neuron latencies (rounding up) to make the analysis robust to a small number of false positives. Area latencies were compared by testing whether the difference between their latencies was significantly different from that expected by chance ($p < 0.05$, permutation test, conducted by comparing the latency difference calculated on the true data to the distribution of latency differences calculated on 20000 permuted data sets which shuffled the assignment of neurons between the two areas being compared).

**Analysis of rough vs. graded uncertainty coding.** In standard uncertainty tasks, a neuron's rough uncertainty activity was calculated as the difference in normalized activity between the pooled data from all uncertain reward CSs and pooled data from the certain 0 and 100% reward CSs. Its graded uncertainty activity was calculated as the difference in normalized activity between data from the uncertain 50% reward CS and pooled data from the uncertain 25 and 75% reward CSs. Neurons were classified as having significant graded uncertainty coding if their graded uncertainty activity was significantly different from 0 ($p < 0.05$, rank-sum test). Areas were classified as having graded uncertainty coding if the number of neurons with significant graded coding was significantly different from chance levels ($p < 0.05$, binomial test).

**Analysis relating neural activity to gaze state.** Gaze was defined as being on the stimulus if it was within a small circular window (3° radius) centered on the trial's CS location. We observed that there was modest but noticeable variance in eye tracker calibration from session to session. For instance, in some sessions the measured gaze location was consistently slightly to the left of the CS at all CS locations, while in other sessions it was slightly to the right. To correct for this, for each neuron and for each CS location, we centered the gaze window on the peak of a smoothed 2D histogram of eye positions collected from all milliseconds when the eye was within an 8° radius of the theoretical center of the CS (using gaze data from all trials that presented a reward-associated CSs and all times from 0.2 s after CS onset to 0.1 s after CS offset). This produced a good match between the gaze window used for analysis and the typical gaze positions around each CS location.

In our first analysis, we tested whether neural activity in each condition (defined as each combination of CS and cue) was altered depending on whether the gaze was on the CS, in a 0.5 -s time window immediately before the animal was going to be informed of the outcome. For the standard uncertainty tasks, this was a pre-outcome window. For the information tasks, this was a pre-cue window on Info trials, and a pre-outcome window on Noinfo trials; the results were calculated separately for these two windows and then averaged. For each neuron and each condition, we did the following procedure. We calculated the neuron's mean normalized activity in the time window at one millisecond resolution, separately for each gaze state (on or off). We then found all time points at which there was valid data for both gaze states (i.e., milliseconds where there was at least one trial with gaze on and one trial with gaze off). We then calculated the mean normalized activity for each time point separately for the "gaze on" and "gaze off" states, and then averaged across time points. This resulted in two measurements for each condition: the neuron's average activity when gaze was on the CS, and its average activity at the same time points when the gaze was off the CS. This analysis was

performed on all neurons where there was trial-to-trial variance in gaze behavior before receipt of information (i.e., at least one millisecond in the time window in which there was at least one trial with gaze on and one trial with gaze off) for all of the CSs. We then tested whether the mean difference in activity between the two gaze states was significantly different from 0 across the neural population ($p < 0.05$, signed-rank test). Note that this analysis essentially asks how neural activity differs across behavioral gaze states. We could have done an equivalent analysis in terms of the reverse relationship, by asking how well gaze states can be predicted from neural activity (e.g., in terms of decoding accuracy). However, our hypothesis was not that these neural populations have a 1:1 relationship with gaze states. Our hypothesis was that neural activity is linked to the level of motivation to seek information, which is one of many factors that compete to influence gaze (other motivational, attentional, perceptual, and motor factors include: expectation of juice reward, general arousal, task engagement, time since visual stimulus onset, perceived stimulus intensity, recent history of saccades, etc.). Therefore, we found it more interpretable to express the neural-behavioral link in terms of modulations of neural activity.

**Analysis relating neural activity to gaze shifts.** We measured the time course of the change in neural activity aligned at the time of gaze shifts. We detected all gaze shifts onto or away from the CS/cue stimulus using the following procedure. Gaze shifts off of the stimulus were defined as milliseconds where (1) the gaze started in a stable "on" state, by being "on" for at least 150 consecutive ms, (2) starting at the current millisecond, the gaze transitioned to a stable "off" state, by being "off" for at least 100 consecutive ms, (3) the gaze was not merely hovering near the edge of the stimulus, indicated by the gaze 100 ms after the putative shift being at a location at least 5° away from the center of the stimulus, (4) the putative gaze shift was not influenced by changes in the eye tracker signal related to blinks, indicated by blinks not occurring in a 100 ms time window before the putative gaze shift. The results were not sensitive to the detailed settings of these parameters; similar results were obtained with other parameters. Gaze shifts onto the stimulus were defined in the analogous manner, except that to make our analysis of pre-shift activity more conservative we subtracted 40 ms from the gaze shift time, so that the gaze shift time more closely approximated the time when the eye movement began rather than the time when the eye entered the gaze window (Fig. 5c, note that time = 0 is approximately when the gaze distance to the stimulus begins to change, rather than when the eye enters the gaze window).

In order to compare neural activity around the time of gaze shifts to neural activity in similar conditions where no gaze shift occurred, we matched each gaze shift with a set of control non-shift events. Gaze shifts were only included in the analysis if at least one control non-shift event was found. For each gaze shift, its control non-shift events were defined as all time points in the data that met the following criteria: (1) were recorded from the same neuron, (2) were on trials that presented the same CS, (3) were on trials that presented the same type of cue, if it was a task that used cues, (4) were at the same time point in the trial as the gaze shift, (5) had spent the previous 150 consecutive ms in the same initial gaze state as the gaze shift, (6) instead of shifting, the gaze continued to stay in that same gaze state for at least 100 consecutive ms. We then selected a subset of these non-shift events to use as controls in our analysis, in order to match the mean gaze-to-stimulus distance before the non-shifts as closely as possible to the gaze-to-stimulus distance before the gaze shift. To do this, we calculated the difference between the gaze-to-stimulus distance 50 ms before each non-shift event vs. 50 ms before the gaze shift. We then selected non-shift events with a procedure that yielded differences that were small and centered around zero. Specifically, we first selected the non-shift event that had the smallest difference. We then iteratively added non-shift events by repeatedly selecting the event whose difference met the following criteria: (A) had the opposite sign of the mean difference of the currently selected events, and (B) had the smallest magnitude of all remaining non-shift events meeting that criterion. For instance, if the first selected non-shift event occurred when gaze was slightly further away from the stimulus than before the gaze shift, the second selected non-shift event would occur when the gaze was slightly closer to the stimulus than before the gaze shift. If no remaining non-shift events met those criteria, no further non-shift events were selected. This procedure yielded 36306 gaze shifts, with a mean of 3.5 control non-shift events per gaze shift (standard deviation: 3.0, range: 1–31). There was a close match between the mean time course of the gaze-to-stimulus distances before gaze shifts vs. before the control non-shift events (Fig. 5c).

Each neuron's activity related to each of its individual gaze shifts was quantified as its normalized activity aligned on the gaze shift minus its mean normalized activity aligned on that gaze shift's associated control non-shift events. The neuron's overall activity related to gaze shifts was quantified by averaging its activity for all individual gaze shifts, separately for each type of gaze shift and each condition being analyzed (e.g., Fig. 5d had two types of gaze shifts: on and off; and three conditions: gaze shifts when the trial's reward outcome was uncertain, known to be reward, or known to be no reward). Neurons were only included in an analysis if they had at least one gaze shift of each type in each of the conditions being compared (e.g., Fig. 5d only includes neurons that had at least one gaze shift in each of the $2 \times 3 = 6$ combinations of gaze shift type and condition). Activity around the time of the gaze shift was quantified in three time windows relative to the gaze shift: before the gaze shift ($-0.4$ to $-0.1$ s), during the gaze shift ($-0.1$ to

+ 0.1 s), and after the gaze shift ( + 0.1 to + 0.4 s). The first two time windows contain activity that cannot be explained as a result of visual feedback following the gaze shift, as the uncertainty-related neurons in these areas almost exclusively had visual response latencies > 0.1 s (e.g., Fig. 1).

**Model of gaze-modulation**. Each neuron's activity during the CS and cue periods in each task was fit with a computational model. The model's parameters were divided into three groups (see Supplementary Fig. 8 for further explanation and illustrations). First, $\beta$ parameters specifying the neuron's responses to CSs and cues: its time course of activity in each task condition during moments when gaze is off the stimulus. This is similar to a traditional PSTH, and represents the time course that neural activity would have if there was no gaze-modulation. Second, $\mu$ and $\sigma$ parameters specifying the time course of gaze-modulation: what times activity should be modulated relative to a gaze shift onto the stimulus, and how strongly gaze should be modulated at those times. Third, $w_{gain}$ and $w_{offset}$ parameters specifying the effect of gaze-modulation: whether gaze induces a multiplicative change in response strength (gain change), an additive increase in firing rate (offset), or both. Importantly, we validated the model by confirming that it accurately recovered the true gaze-modulation parameters when it was fitted to simulated data sets, including simulations where (A) gaze-modulation effects were similar to those in the real data, (B) gaze-modulations had the same time course as the real data but a variety of different magnitudes, (C) gaze-modulations had the same magnitudes as the real data but a variety of different time courses, (D) "null hypothesis" simulations in which gaze-modulations were absent (Supplementary Fig. 8).

The model was defined as follows. The $\beta$ parameters specified the mean firing rate when gaze was off the stimulus location. A separate parameter, $\beta(c,t)$, was defined for each trial condition $c$ and each time bin $t$ during the trial. Thus, similar to a conventional set of PSTHs, they specified the full time course of the neural response to each stimulus. Trial conditions were defined as combinations of CSs and cues. For instance, in the information task shown in Fig. 2 there were seven conditions (Info 100% CS with reward cue, Info 50% CS with reward cue, Info 50% CS with no-reward cue, Info 0% CS with no-reward cue, Noinfo 100% CS with noninformative cue, Noinfo 50% CS with noninformative cue, Noinfo 0% CS with noninformative cue). Time bins were defined as 50 ms bins spanning from 0.2 s after CS onset to 0.1 s after outcome delivery onset. Thus, a neuron recorded with a 3 s total duration from CS onset to outcome delivery was fit with seven conditions × 58 time bins = 406 distinct $\beta$ parameters.

The $\mu$ and $\sigma$ parameters specified the time course of gaze-modulation (Supplementary Fig. 8). Specifically, at each millisecond of each trial, a GazeMod variable specified the degree to which neural activity was modulated by gaze (0 <= GazeMod <= 1, 0 = no modulation, 1 = maximal modulation). GazeMod was computed by convolving a binary Gaze variable (1 = gaze is on the stimulus, 0 = gaze is off the stimulus) with a Gaussian kernel with mean = $\mu$ and standard deviation = $\sigma$. Thus, $\mu$ controlled the temporal offset between neural activity and gaze (whether neural activity is modulated before the gaze shifts onto the stimulus, or vice versa) and $\sigma$ controlled the gradualness of gaze-modulation around a gaze shift (low $\sigma$ = rapid onset of gaze-modulation, high $\sigma$ = gradual onset). Thus, the time course of gaze-modulation around the time of a gaze shift took the form of a cumulative Gaussian function (Supplementary Fig. 8). The model's gaze-modulation latency was calculated as the time relative to gaze onset when its time course of gaze-modulation reached 10% of its maximal value (i.e., when GazeMod = 0.1); other criteria produced the same main result (i.e., ACC before icbDS and Pal).

The $w_{gain}$ and $w_{offset}$ parameters specified the effect of gaze-modulation (Supplementary Fig. 8). Specifically, $w_{gain}$ caused gaze to multiplicatively scale firing rates, while $w_{offset}$ caused gaze to add or subtract from firing rates. Putting the parameters together, the neural firing rate in time bin $t$ of trial $tr$ that was in condition $c$ was modeled as:

$$\text{Rate}(tr,t) = \beta(c,t) \times \left(1 + \text{GazeMod}(tr,t) \times w_{gain}\right) + \text{GazeMod}(tr,t) \times w_{offset} + \varepsilon \quad (2)$$

…where GazeMod(tr,t) is the mean GazeMod on trial $tr$ in time bin $t$, and $\varepsilon$ is a normally distributed noise term. Thus, if $w_{gain} = w_{offset} = 0$ then there was no gaze-modulation. If one of these parameters was nonzero, then gaze had a strictly multiplicative or additive effect on firing rate. Finally, if both parameters were nonzero, then the model multiplicatively increased the gain of neural responses relative to a baseline firing rate (e.g., during gaze, excitations above baseline are stronger excitations, and inhibitions below baseline are stronger inhibitions; Supplementary Fig. 8).

The model was fitted using the method of maximum likelihood, i.e., finding the parameters that maximize the probability that the model would produce the observed data. We did this by deriving the log likelihood function, its gradient, and its Hessian, and using them as input to the Trust Region Reflective Algorithm for function optimization to find the parameters that locally maximized the log likelihood (using its implementation in the fmincon function in Matlab). The parameters were constrained so that $\mu$ was in $[-1, +1]$ and $\sigma$ was in $[0,1]$; the other parameters were unconstrained. The initial parameter settings were $\beta = 0$, $\mu = 0$, $\sigma = 0.1$, $w_{gain} = 0$, $w_{offset} = 0$; different initial parameter settings gave similar results. The optimization algorithm was allowed to continue for each session (here, defined as each set of data collected from a particular neuron with a particular task)

for up to 100 function evaluations. The optimization algorithm successfully converged on 459/460 (99.8%) of sessions.

A neuron was considered to be significantly modulated by gaze if the log likelihood of the gaze model was significantly higher than expected by chance under the null hypothesis that there was no relationship between neural activity and gaze ($p < 0.05$, permutation test with 200 permutations). This was tested by comparing the log likelihood of the model fitted to the neuron's true data to the distribution of log likelihoods of the model fitted to shuffled data sets in which the neural data was exactly the same as the original data set, but the gaze data were randomly shuffled among trials that shared the same task condition. For instance, suppose the 75% reward CS was presented on trials 5, 12, 14, 22, and 24. The neural data on those trials would be kept the same, but the gaze data would be shuffled so that the gaze data from trial 5 might now be assigned to trial 22; the gaze data from trial 12 might now be assigned to trial 14; etc. Thus, shuffling destroyed any trial-to-trial relationships between neural activity and gaze, while leaving the neural and gaze data individually intact. The neuron was classified as significantly gaze-modulated if the log likelihood of the model fit to the true data set was greater than the log likelihoods of the model fits to at least 191/200 of the permuted data sets. Note that this is a one-tailed test, because the shuffling cannot (in expectation) improve the quality of the fit. That is, if there was a true neural-gaze relationship in the original data then shuffling should worsen the fit, while if there was no such relationship then shuffling should not affect the quality of the fit; in neither case could shuffling improve the fit.

The latency of gaze-modulation was compared across areas using two methods which gave consistent results. First, we calculated each area's mean time course of gaze-modulation by averaging the fitted time courses for each neuron in that area with significant gaze-modulation (Fig. 6f) and using this to calculate the area's latency in the same method used for single-neuron time courses. We then calculated whether the per-area latencies were significantly different ($p < 0.05$, permutation test conducted by comparing the difference between the area latencies with the distribution of such differences computed from 2000 permuted data sets, in which the assignment of neurons to the two areas was randomly shuffled). Second, we directly compared the distribution of fitted latencies from all single neurons with significant gaze-modulation in the two areas (Fig. 6f inset, rank-sum test, $p < 0.05$).

**Analysis of perturbation experiments**. Response times (RTs) were computed online and used to determine when the CS was replaced by the cue, defined as the time between the go signal (i.e., the trial start cue's disappearance) and the gaze entering the response window around the CS. To improve the accuracy of RT measurements for our offline analysis, RTs were recomputed offline using response windows that were corrected for session-to-session variability in eye tracker settings using the procedure described above (i.e., centering the window on the observed peak gaze location separately for each CS location and each session). We then analyzed the RTs from all correctly performed trials in which the animal made a response and there was at least rough agreement between the online and offline RTs (i.e., within 0.2 s of each other). These criteria were met by nearly all correctly performed trials ($n = 17968/18035$; 99.6%). We then quantified the information-seeking bias using an Infobias Index based on the mean RTs for the Info and Noinfo CSs:

$$\text{Infobias index} = (\text{Noinfo RT} - \text{Info RT})/(\text{Noinfo RT} + \text{Info RT}), \quad (3)$$

…which was computed separately for each session, and separately for each of the 2 × 2 combinations of time in session (pre- vs. post-injection) and CS location relative to injection site (contralateral vs. ipsilateral). We then derived two additional measures. For session and each CS location, we defined the change in Infobias index as the difference between post-injection and pre-injection Infobias indexes. We defined the change in Infobias laterality as the difference between the changes in Infobias Index for the contralateral and ipsilateral sides.

To test whether icbDS and Pal inactivations affected information-seeking behavior (Fig. 7c), we computed the mean change in Infobias index and tested whether it was significantly different from zero ($p < 0.05$, permutation test conducted by comparing the true mean change in Infobias Index to the distribution of mean changes in Infobias Index computed on 20,000,000 permuted data sets, in which pre- and post-injection data were shuffled with each other). For pooled data from all inactivation sessions and for control sessions (Fig. 7c), we used the same procedure, except to be conservative we included an additional correction for any potential main effects of animal, by using weighted means such that each animal's inactivation and control data were weighted by the number of inactivation sessions that animal contributed to the data set. The same key results were obtained in uncorrected data (significant change in contralateral Infobias Index during inactivation sessions, $p = 0.0015$, signed-rank test; no significant change in Control sessions, $p = 0.8786$, signed-rank test; changes in inactivation sessions significantly different from changes in control sessions, $p = 0.0297$, rank-sum test).

To test how inactivations interfered with information-seeking behavior, we analyzed RTs separately for Info and Noinfo CSs. RTs were normalized by z-scoring all RTs separately for each session and CS location. Then for each area and each CS type, we tested whether there was a significant difference between the pre- vs. post-injection RT distributions for contralateral CSs ($p < 0.05$, ranked-sum test). We further tested if the changes in RTs were significantly different for the two CS

types ($p < 0.05$, interaction term from a two-way ANOVA using the factors CS type (Info or Noinfo) and epoch in session (pre- or post-injection)). Finally, to directly compare the effects of injections in the two areas, we first computed each area's overall change in mean normalized RT averaged across the Info and Noinfo CSs. This was computed as follows:

$$\Delta \mathrm{RT}_{\mathrm{area,overall}} = 0.5 \times \left( \mathrm{RT}_{\mathrm{area,Info,after}} - \mathrm{RT}_{\mathrm{area,Info,before}} \right) + 0.5 \times \left( \mathrm{RT}_{\mathrm{area,Noinfo,after}} - \mathrm{RT}_{\mathrm{area,Noinfo,before}} \right),$$
(4)

where $\mathrm{RT}_{\mathrm{area,condition}}$ is the mean normalized RT from experiments with a given area and in the specified condition (Info or Noinfo CS and before or after injection). We tested whether $\Delta \mathrm{RT}_{\mathrm{icbDS,overall}}$ was significantly different from $\Delta \mathrm{RT}_{\mathrm{Pal,overall}}$ (permutation test, conducted by comparing the difference between the $\Delta \mathrm{RT}_{\mathrm{overall}}$ of the two areas and comparing it to the distribution of differences computed on 2000 permuted data sets in which each session's pre- and post-injection data were shuffled with each other). We then computed the component of this overall change in RTs that occurred in the hypothesized manner, i.e., icbDS injection slowing RTs to the Info CS or Pal injection speeding RTs to the Noinfo CS, as:

$$\Delta \mathrm{RT}_{\mathrm{hypothesized}} = \left( \mathrm{RT}_{\mathrm{icbDS,Info,after}} - \mathrm{RT}_{\mathrm{icbDS,Info,before}} \right) - \left( \mathrm{RT}_{\mathrm{Pal,Noinfo,after}} - \mathrm{RT}_{\mathrm{Pal,Noinfo,before}} \right),$$
(5)

We also computed the component that occurred in the orthogonal manner to our hypothesis, i.e., icbDS injection slowing RTs to the Noinfo CS or Pal injection speeding RTs to the Info CS, as:

$$\Delta \mathrm{RT}_{\mathrm{orthogonal}} = \left( \mathrm{RT}_{\mathrm{icbDS,Noinfo,after}} - \mathrm{RT}_{\mathrm{icbDS,Noinfo,before}} \right) - \left( \mathrm{RT}_{\mathrm{Pal,Info,after}} - \mathrm{RT}_{\mathrm{Pal,Info,before}} \right),$$
(6)

We tested these components for significance using the same type of permutation test (comparing them to the distribution of the components calculated from 2000 permuted data sets in which each session's pre- and post-injection data were shuffled with each other).

To test how inactivations might be expected to interfere with information seeking given the broad strokes of the cortico-BG circuit, we implemented a simple computational model of the network and used it to generate a simulated RT for each trial in our data set (Fig. 7e). We then analyzed the simulated RTs in the same manner as the real RTs. Note that this model is intended as a simple test of whether information-seeking behavior and its perturbation by icbDS and Pal inactivations are consistent with a straightforward implementation of the typical cortico-BG information signals we observed and the signs of excitatory/inhibitory connections between areas. This model is not intended to emulate the detailed or recurrent dynamics of neural activity within or between areas or to emulate the detailed circuitry of visual processing and saccade generation. In this model, each neural population in each hemisphere was represented by a single simulated neuron whose firing rate $r$ on each trial was:

$$r_{\mathrm{ACC}} = w_{\mathrm{ACC}} + I \times w_{I \to \mathrm{ACC}},$$
(7)

$$r_{\mathrm{icbDS}} = w_{\mathrm{icbDS}} + r_{\mathrm{ACC}} \times w_{\mathrm{ACC} \to \mathrm{icbDS}},$$
(8)

$$r_{\mathrm{Pal}} = w_{\mathrm{Pal}} + r_{\mathrm{icbDS}} \times w_{\mathrm{icbDS} \to \mathrm{Pal}},$$
(9)

$$r_{\mathrm{Visual}} = w_{\mathrm{Visual}} + S \times w_{S \to \mathrm{Visual}},$$
(10)

$$r_{\mathrm{Gaze}} = w_{\mathrm{Gaze}} + r_{\mathrm{Visual}} \times w_{\mathrm{Visual} \to \mathrm{Gaze}} + r_{\mathrm{Pal}} \times w_{\mathrm{Pal} \to \mathrm{Gaze}} + \varepsilon,$$
(11)

…where for each area X, the variable $w_X$ represents the area's baseline firing rate and $w_{Y \to X}$ represents the weight of incoming input from area Y to area X in the same hemisphere, $I$ represents the availability of information and is 1 for trials with an Info CS and 0 for trials with a Noinfo CS, $S$ represents the presence of a contralateral visual stimulus, and is 1 for trials with a contralateral CS and 0 for trials with an ipsilateral CS, and $\varepsilon$ is Gaussian noise drawn independently on each trial with mean = 0 and standard deviation = 30. Here, Visual represents a population of neurons responsive to visual input for the purpose of directing eye movements (e.g., visually responsive neurons in the frontal eye field, lateral intraparietal sulcus, and/or the superior colliculus[105–107]), and Gaze represents a population of neurons that directly control eye movements (e.g., saccade-related neurons in the deep layers of superior colliculus[105]). Note that the connections in this model are not meant to imply that effects are necessarily mediated in the brain by direct monosynaptic connections; the connections correspond to the net effect of both direct and indirect influences of activity in one area on another area. The baseline and input weights of each cortico-BG area were selected to approximately match the typical firing rates of information-related neurons in that area in the moment before cue delivery in the information task, and to be consistent with the excitatory/inhibitory nature of each connection in the cortico-BG network, as follows. Baseline rates were: ACC = 2, icbDS = 0, Pal = 50, Visual = 0, Gaze = 0. Input weights were: Info → ACC = + 20, Stim → Visual = + 100, ACC → icBDS = + 1, icbDS → Pal = −2, Visual → Gaze = + 1, Pal → Gaze = −1. Finally, the simulated RTs for each trial were generated by taking the firing rates from the Gaze neuron in the contralateral hemisphere to the CS and mapping them onto the distribution of RTs in the real data (so that the trial with the Kth highest firing rate from the Gaze neuron was given the Kth fastest RT observed in control sessions in the real data). These simulated RTs were then analyzed in the same manner as the real data. Inactivations were simulated by multiplying the firing rate of the inactivated area (icbDS or Pal) by a scaling factor of 0.7, representing a 30% reduction in firing rate; other scaling factors produced qualitatively similar results (i.e., icbDS inactivations predominantly slowing Info CS RTs and Pal inactivations predominantly speeding Noinfo CS RTs).

**Anatomy and tracer injections**. Experiments were carried out on separate animals from those used for electrophysiology. Some of these injections were analyzed for other anatomical studies of the striatal-pallidal network (e.g., ref. [26,108]). Procedures were conducted in accordance with the Institute of Laboratory Animal Resources Guide for the Care and Use of Laboratory Animals and approved by the University Committee on Animal Resources at the University of Rochester. Adult male macaque monkeys (data from 1 *Macaca nemestrina*, 2 *Macaca fascicularis* shown here) were tranquilized by intramuscular injection of ketamine (10 mg/kg). For a subset of animals, MRI (3 T) T1 or T2 turbo spin echo scans ($0.5 \times 0.5 \times 1.42$ mm) were obtained before surgery. For the others, serial electrode penetrations were made to locate the anterior commissure: neuronal activity was identified based on patterns of electrophysiological activity of the striatum, both segments of the pallidum, the ventral pallidum, and the underlying nucleus basalis, while the absence of cellular activity was used to distinguish the anterior commissure from the external pallidal segment dorsally and the subcommissural ventral pallidum ventrally[109]. These images and recordings were used to calculate the anterior/posterior, dorsal/ventral, and medial/lateral coordinates for each tracer injection from stereotaxic zero. Animals received ketamine 10 mg/kg, diazepam 0.25 mg/kg, and atropine 0.04 mg/kg IM in the cage. A surgical plane of anesthesia was maintained by either intravenous injections of pentobarbital (for recordings, initial dose 20 mg/kg, i.v., and maintained as needed) or via 1–3% isoflurane in 100% oxygen via vaporizer. Temperature, heart rate, and respiration were monitored throughout the surgery. Monkeys were placed in a Kopf stereotaxic, a midline scalp incision was made, and the muscle and fascia were displaced laterally to expose the skull. A craniotomy (~2–3 cm$^2$) was made over the region of interest, and small dural incisions were made only at injection sites. Monkeys received an injection of one or more of the following anterograde/bidirectional tracers: lucifer yellow, fluororuby, or fluorescein conjugated to dextran amine (LY, FR, or FS; 40–50 nl, 10% in 0.1 M phosphate buffer (PB), pH 7.4; Invitrogen); wheat germ agglutinin conjugated to horseradish peroxidase (WGA; 40–50 nl, 4% in distilled water; Sigma, St. Louis, MO); *Phaseolus vulgaris*-leucoagglutinin (PHA-L; 50 nl, 2.5%; Vector Laboratories); or tritiated amino acids (AA, 100 nl, 1:1 solution of [$^3$H] leucine and [$^3$H]-proline in dH2O, 200 mCi/ml, NEN). These tracers do not cross-react with one another, and thus an individual animal can serve in multiple experiments, reducing the total number of animals needed. Tracers were pressure-injected over 10 min using a 0.5-μl Hamilton syringe. Following each injection, the syringe remained in situ for 20–30 min. Twelve to 14 days after surgery, monkeys were again deeply anesthetized and perfused with saline followed by a 4% paraformaldehyde/1.5% sucrose solution in 0.1 M PB, pH 7.4. Brains were postfixed overnight, and cryoprotected in increasing gradients of sucrose (10, 20, and 30%). Serial sections of 50 μm were cut on a freezing microtome into 0.1 M PB or cryoprotectant solution[110]. One in eight sections was processed free-floating for immunocytochemistry to visualize the tracers. Tissue was incubated in primary anti-LY (1:3000 dilution; Invitrogen) or anti-FR (1:1000; Invitrogen in 10% NGS and 0.3% Triton X-100 (Sigma- Aldrich) in PB for four nights at 4 °C. Following extensive rinsing, the tissue was incubated for 40 min in biotinylated secondary anti-rabbit antibody made in goat (1:200; Vector BA-1000) followed by incubation with the avidin–biotin complex solution (Vectastain ABC kit, Vector Laboratories). Immunoreactivity was visualized using standard DAB procedures. Staining was intensified by incubating the tissue for 5–15 s in a solution of 0.05% 3,3′-diaminobenzidine tetra-hydrochloride, 0.025% cobalt chloride, 0.02% nickel ammonium sulfate, and 0.01% H2O2. Sections were mounted onto gel-coated slides, dehydrated, defatted in xylene, and cover-slipped with Permount. Using darkfield light microscopy, brain sections, injection sites, and dense pallidal terminal fields were outlined under a 1.6, 4.0, or 10 × objective using a Leitz or Leica microscope with Neurolucida software (MBF Bioscience). Terminal fields were considered dense when they could be visualized at a low objective (1.6 ×)[110]. Retrogradely labeled input cells were identified under brightfield microscopy (×20). StereoInvestigator software (Micro-BrightField) was used to stereologically count labeled cells with an even sampling (64%). On the additional three cases (45LY, 113FS, 40LY), cells were charted in representative select frontal sections using the same parameters. Other sections were visually inspected for labeling.

**Reporting summary**. Further information on research design is available in the Nature Research Reporting Summary linked to this article.

## Data availability
The data sets generated during and/or analyzed during the current study are available from the corresponding author on reasonable request.

## Code availability
The custom code for model fitting is available from the corresponding author on reasonable request.

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

## Acknowledgements

We are grateful to Dr. Noah Ledbetter for assisting in data acquisition, Michael Traner for assisting with pharmacology experiments, and Ms. Kim Kocher for fantastic animal care and training. This work was supported by the National Institute of Mental Health under Award Numbers R01MH110594 and R01MH116937 to I.E.M., and by the McKnight Memory and Cognitive Disorders Award, the NARSAD award, and the Edward Mallinckrodt, Jr. Foundation award to IEM. Also, we wish to extend many thanks to the McDonnell Center for Systems Neuroscience for the initial seed support for this work.

## Author contributions

J.K.W. contributed to study design, led data collection for electrophysiology and pharmacology, and contributed to data analysis; E.S.B.-M. contributed to study design and led data analysis and writing the paper; S.R.H. performed the anatomical analysis; K.Z. contributed to data collection for electrophysiology and pharmacology and data analysis; J.P. contributed to data collection for electrophysiology; S.N.H. provided anatomical data and supported the anatomical analysis; I.E.M. acquired the funding for this work, contributed to conceptualization and study design, and supervised the data collection, analyses and writing of this paper.

## Competing interests

The authors declare no competing interests.

## Additional information

**Supplementary information** is avalilble for this paper at https://doi.org/10.1038/s41467-019-13135-z.

