## [Peer Review File · Nature Communications]

Editorial Note: This manuscript has been previously reviewed at another journal that is not operating a transparent peer review scheme. This document only contains reviewer comments and rebuttal letters for versions considered at *Nature Communications*. Mentions of the other journal have been redacted.

Reviewers' comments:

Reviewer #1 (Remarks to the Author):

The authors have addressed all my questions and concerns.

Reviewer #3 (Remarks to the Author):

I have provided an assessment of this work as Reviewer #3 at [Redacted]. The authors have addressed my remarks in a thorough and satisfactory fashion. They now discuss open/alternative interpretations of their findings, and have added supplementary material from an additional control condition to support one of their claims. They also relate their findings to prominent competing theories regarding the function of the ACC in human learning and decision-making.

I am also happy with the authors' response regarding my methodological point about their brain-behavior analyses. The added statements in the Methods now state explicitly why the authors have conducted their analysis in a direction (i.e., assessing modulations of neural activity as a function of behavioral variables) that is opposite to the implicit causal chain of events (that is, the fact that it is neural activity that produces behavior rather than the opposite).

I do not have further concerns regarding the current version of the manuscript, and congratulate the authors for a very interesting study.

Valentin Wyart

Reviewer #4 (Remarks to the Author):

Review for monkey "a neural network for information seeking" by White, Bromberg-Martin and co.

The authors investigated the neural bases of information seeking using a combination of electrophysiology, neuro-anatomy and pharmacological manipulation of neural activity. I was not one of the reviewers of the first submission [Redacted] but I took the time to read both the article and the rebuttal letter. I found the article very thorough and their responses to the (very fair) reviewers' concern exhaustive. The authors should be particularly praised for recording additional neurons (94 monkey B), running simulations to assess the statistical power of their main conclusions and showing the additional results and analyses of the condition where reward is contingent upon action.

I have just two remaining questions that I would like to be addressed. One concerning the analyses and the other concerning the discussion.

1. The authors show that the normalized activities in the three regions of interest is higher in the uncertainty conditions (25/50/75) compared to the certain ones (0/100). They show this in Figure 1. They also claim that the response is "graded" as response to 50 (maximal uncertainty) is higher compared to the 25/75 conditions. This is not very clear from the data and the analysis. The curve (Figure 1E) is almost flat from the 25 to 75 conditions. The claim is based on a subset of neurons (~20%?) displaying a significant effect (even though the overall distributions seems centered at zero). Since this result (graded response) is central for the conclusions, it would be nice to see it confirmed in a more formal manner, for example comparing a quadratic model to a step model and providing more statistics (for example for neurons whose response for $50 > 25/75$, is it true comparing to both 25 and 75 separately?).

2. The discussion is a bit stunted in respect of the relation of these results to well known results about uncertainty encoding by the dopaminergic and noradrenergic modulatory systems, which project heavily in the investigated structures. It is important that the authors discuss how their results fit this literature and what pharmacological perturbation of these two systems would cause to these neural signals.

Reviewer #1 (Remarks to the Author):

The authors have addressed all my questions and concerns.

Response: We thank reviewer 1 for their comprehensive, detailed comments.

Reviewer #3 (Remarks to the Author):

I have provided an assessment of this work as Reviewer #3 at [Redacted]. The authors have addressed my remarks in a thorough and satisfactory fashion. They now discuss open/alternative interpretations of their findings, and have added supplementary material from an additional control condition to support one of their claims. They also relate their findings to prominent competing theories regarding the function of the ACC in human learning and decision-making.

I am also happy with the authors' response regarding my methodological point about their brain-behavior analyses. The added statements in the Methods now state explicitly why the authors have conducted their analysis in a direction (i.e., assessing modulations of neural activity as a function of behavioral variables) that is opposite to the implicit causal chain of events (that is, the fact that it is neural activity that produces behavior rather than the opposite).

I do not have further concerns regarding the current version of the manuscript, and congratulate the authors for a very interesting study.

Valentin Wyart

Response: We thank reviewer 3 for their insightful feedback, especially about clarifying our methods and relating our results to relevant literature.

Reviewer #4 (Remarks to the Author):

Review for monkey "a neural network for information seeking" by White, Bromberg-Martin and CO.

The authors investigated the neural bases of information seeking using a combination of electrophysiology, neuro-anatomy and pharmacological manipulation of neural activity. I was not one of the reviewers of the first submission [Redacted], but I took the time to read both the article and the rebuttal letter. I found the article very thorough and their responses to the (very fair) reviewers' concern exhaustive. The authors should be particularly praised for recording additional neurons (94 monkey B), running simulations to assess the statistical power of their main conclusions and showing the additional results and analyses of the condition where reward is contingent upon action.

Response: We thank reviewer 4 for their careful read of our original and new manuscripts and of our reply! We are pleased by their assessment that our article is thorough and addresses the reviewer feedback.

I have just two remaining questions that I would like to be addressed. One concerning the analyses and the other concerning the discussion.

1. The authors show that the normalized activities in the three regions of interest is higher in the uncertainty conditions (25/50/75) compared to the certain ones (0/100). They show this in Figure 1. They also claim that the response is “graded” as response to 50 (maximal uncertainty) is higher compared to the 25/75 conditions. This is not very clear from the data and the analysis. The curve (Figure 1E) is almost flat from the 25 to 75 conditions. The claim is based on a subset of neurons (~20%?) displaying a significant effect (even though the overall distributions seems centered at zero). Since this result (graded response) is central for the conclusions, it would be nice to see it confirmed in a more formal manner, for example comparing a quadratic model to a step model and providing more statistics (for example for neurons whose response for $50 > 25/75$, is it true comparing to both 25 and 75 separately?).

Response: These are good points! The reviewer correctly points out that the response to 50 is only modestly higher than the responses to 25 and 75. Perhaps surprisingly, this is actually exactly the expected result if neurons truly encode the graded level of uncertainty! This is because the predominant measures of uncertainty that have been proposed in the literature, such as entropy and standard deviation, are inverted-U functions of reward probability that are only modestly higher for 50 than 25 and 75 (as illustrated by the theoretical curves plotted below). Hence neurons encoding uncertainty would be expected to show the inverted-U pattern $50 > 25, 75 \gg 0$ much like that seen in our data.

In addition, we would like to emphasize that our statistical measurements of graded coding are *not* simply based on neurons with a significant $50 > 25/75$ effect (which the reviewer raises as a potential concern). We realize that did not explain our analysis well enough to make this clear!

Of course, as the reviewer correctly points out, if we had only analyzed the neurons consistent with our hypothesis then that would be an invalid analysis rife with ‘double dipping’ selection bias: even if a distribution is truly centered at zero, ‘chopping off’ the right side of the distribution and only analyzing those neurons would give the false impression of an impressive effect.

Instead, we avoid selection bias by using an unbiased approach: our key statistics are based on including *all* neurons that had significant differential coding between 50 vs 25/75, regardless of whether they did so in the direction consistent with our hypothesis of graded coding ($50 > 25/75$) or in the opposite direction that goes against our hypothesis ($50 < 25/75$).

This analysis led to two key findings.

First, we found that 19% of neurons had significant differential signals ($p < 0.05$, signed-rank test). This is far greater than the chance level of 5% that would be expected under the null hypothesis that neurons encode all levels of uncertainty in a ‘flat’, non-differential manner ($p < 0.0001$, binomial test). Thus, we can reject the null hypothesis that neurons encoded uncertainty in a flat manner.

Second, we found that of these neurons with significant differential coding, fully 97% of them did so in the direction consistent with our hypothesis of graded coding (dark blue cells in Fig 1F, $50 > 25/75$)! Only the remaining 3% of them did so in the opposite direction (cyan cell in Fig 1F, $50 < 25/75$). Of course, there is a significant difference between the number of neurons doing the former vs. the latter ($p < 0.0001$, binomial test). Thus, the differential coding neurons are overwhelmingly consistent with graded coding.

That said, we can see how this could be confusing to readers! When readers see Fig 1F, they might think that we are *only* plotting the neurons that favor our hypothesis and ignoring those with the opposite pattern...when in fact we are plotting *all* differential neurons, and it is simply the case that 97% of these neurons are on the right side of the plot that favors graded coding, while barely any neurons have the opposite pattern.

We have now revised our manuscript to address this point by adding further description of the current statistical findings, as well as doing additional model fitting and tests of neuronal data following the reviewer’s further suggestions. Specifically,

1. We have clarified our text to emphasize that the tests using the subset of neurons with significant activity differences are not prey to the concerns described above. The new text reads:

“Specifically, neurons with a significantly different average response to the 50% CS compared to the 75 and 25% CSs were found in all three areas (Fig 1F, solid colors, $p < 0.05$, signed-rank test) at a much greater prevalence than expected by chance (19% of neurons ($n=32/165$); $p < 0.001$ in each area, binomial tests). Of these neurons, 97% had

significantly stronger responses to maximal uncertainty than to intermediate uncertainty, consistent with graded uncertainty coding (Fig 1F, n=31, dark blue), while only 3% had the opposite pattern (significantly stronger responses to intermediate uncertainty than to maximal uncertainty; Fig 1F, n=1, cyan). As a result, there were significantly more neurons with responses consistent with graded uncertainty coding than neurons with the opposite pattern (Fig 1F; $p < 0.001$ in each area, binomial tests) and the average differential activity was greater for maximal uncertainty than intermediate uncertainty ($p < 0.01$ in each area, signed-rank tests)."

2. Following the reviewer's suggestions for additional analysis, we now include a new Supplementary Note 1 with additional discussion of the above points and additional statistical tests of (A) whether uncertainty coding activity had both $50 > 25$ and $50 > 75$ in specific comparisons between those individual CSs, (B) a formal model comparison between models in which uncertainty is encoded in a graded, inverted-U function of reward probability, or is encoded as a simple binary step function. Both of these tests produced the expected results in line with our conclusions.

Notably, as further evidence that our results are not affected by selection bias, these analyses produced the same conclusions when run using either the subset of neurons with significant differential coding of 50 vs 25/75, or when run using *all* uncertainty coding neurons in the dataset regardless of their differential coding status.

2. The discussion is a bit stunt in respect of the relation of these results to well known results about uncertainly encoding by the dopaminergic and noradrenergic modulatory systems, which project heavily in the investigated structures. It is important that the authors discuss how their results fit this literature and what pharmacological perturbation of these two systems would cause to these neural signals.

Response: This is a good point that we are glad to add to our discussion. We have now revised our manuscript to add a discussion of the potential roles of neuromodulators in regulating the network's activity and information seeking behavior, as follows:

"Indeed, information-related neurons in all three areas were intermixed with other neurons that encoded the reward value of stimuli and outcomes, as expected from previous studies of these areas^{13,22,30,39,40,49-52}. This suggests that information- and primary reward-related neurons are well-positioned to support each other's computations. For instance, information-anticipatory activity in our tasks can be interpreted as ramping up to the expected time of a large reward prediction error (due to being informed of a pleasing or disappointing future outcome). Thus, while most uncertainty-related neurons in these areas do not encode reward prediction errors themselves²², their activity could prepare local reward-processing networks to handle upcoming reward prediction error or surprise signals, processes in which ACC, dorsal striatum, and pallidum have been implicated^{40,53-57}. Conversely, the network could learn its tonic information-anticipatory activity by treating reward prediction errors as a teaching signal, such that large prediction errors evoked by surprising outcomes lead to greater ramping on

future trials, while small prediction errors evoked by correctly predicted outcomes reduce ramping on future trials. Indeed, icbDS ramping activity is strong during initial exposure to a novel, ambiguous situation, but rapidly diminishes if animals can learn to correctly predict its reward outcomes²¹. This learning process could be mediated by input from midbrain dopamine neurons, which encode phasic reward prediction errors in response to informative feedback^{13,58}. Other neuromodulators could serve this role as well. The network receives acetylcholine from tonically active neurons in the striatum which respond during reward prediction errors⁵⁹ and from the basal forebrain which contains neurons with activity related to surprise⁶⁰⁻⁶²; the network also receives norepinephrine from the locus coeruleus, which contains neurons with activity related to certain rewarding stimuli and actions⁶³⁻⁶⁵ and is linked to pupil dilation⁶⁶ which can covary with surprise and uncertainty^{67,68}. This raises the possibility that disruption of neuromodulators in the network could impair learning or performance of information seeking behavior.”

REVIEWERS' COMMENTS:

Reviewer #4 (Remarks to the Author):

The authors successfully addressed my questions.